



# Simulating Optical Top-Of-Atmosphere Radiance Satellite Images over Snow-Covered Rugged Terrain

Maxim Lamare[1,2], Marie Dumont[1], Ghislain Picard[2], Fanny Larue[2], François Tuzet[1,2], Clément Delcourt[1,2,3], and Laurent Arnaud[2]

[1]Univ. Grenoble Alpes, Université de Toulouse, Météo-France, CNRS, CNRM, Centre d'Etudes de la Neige, 38000 Grenoble, France
[2]UGA, CNRS, Institut des Géosciences de l'Environnement (IGE) UMR 5001, Grenoble, France
[3]now at: VU University, Faculteit der Bètawetenschappen, cluster Aarde en Klimaat, De Boelelaan 1081-1087, Amsterdam, The Netherlands

**Correspondence:** M. Lamare (maxim.lamare@meteo.fr)

**Abstract.** The monitoring of snow-covered surfaces on Earth is largely facilitated by the wealth of satellite data available, with increasing spatial resolution and temporal coverage over the last years. Yet to date, retrievals of snow physical properties still remain complicated in mountainous areas, owing to the complex interactions of solar radiation with terrain features such as multiple scattering between slopes, exacerbated over bright surfaces. Existing physically-based models of solar radiation

across rough scenes are either too complex and resource-demanding for the implementation of systematic satellite image processing, not designed for highly reflective surfaces such as snow, or tied to a specific satellite sensor. This study proposes a new formulation, combining a forward model of solar radiation over rugged terrain with dedicated snow optics into a flexible multi-sensor tool that bridges a gap in the optical remote sensing of snow-covered surfaces in mountainous regions. The model presented here allows to perform rapid calculations over large snow-covered areas. Good results are obtained even

for extreme cases, such as steep shadowed slopes or on the contrary, strongly illuminated sun-facing slopes. Simulations of Sentinel-3 OLCI scenes performed over a mountainous region in the French Alps allow to reduce the bias by up to a factor 6 in the visible wavelengths compared to methods that account for slope inclination only. Furthermore, the study underlines the contribution of the individual fluxes to the total top-of-atmosphere radiance, highlighting the importance of reflected radiation from surrounding slopes which, in mid-winter after a recent snowfall (13 February 2018), account on average for 7% of

the signal at 400 nm and 16% at 1020 nm (on 13 February 2018), as well as coupled diffuse radiation scattered by the neighbourhood, that contributes to 18% at 400 nm and 4% at 1020 nm. Given the importance of these contributions, accounting for slopes and reflected radiation between terrain features is a requirement for improving the accuracy of satellite retrievals of snow properties over snow-covered rugged terrain. The forward formulation presented here is the first step toward this goal, paving the way for future retrievals.



## 1  Introduction

Seasonal snow plays an essential role in the Earth's climate system owing to its high reflectivity causing strong feedback loops (e.g. Flanner et al., 2011; Qu and Hall, 2006) and thermal properties which impact local energy fluxes between the surface and the atmosphere (Cohen, 1994). In mountainous regions, snow cover spatial and temporal variations have strong environmental and economic implications through, for instance, the control of hydrologic processes such as freshwater storage (Williams et al., 2009) and availability for downstream populations (Barnett et al., 2005), vegetation activity (Trujillo et al., 2012), or natural hazards (Jamieson and Stethem, 2002). Hence, monitoring the properties of the snow cover in mountainous regions is essential to provide policy-makers with reliable information for environmental and societal management, as well as for monitoring climate change.

The benefits of remote sensing techniques to characterise snow have been long-established (e.g. Dietz et al., 2012; Dozier and Painter, 2004; König et al., 2001; Kokhanovsky et al., 2019; Nolin, 2010). Commonly, methods to derive information about the snow physical properties from optical satellite observations are based on surface reflectance products (e.g. Campagnolo et al., 2016; Fily et al., 1997; Klein and Stroeve, 2002; Kokhanovsky and Schreier, 2008; Malcher et al., 2003; Qu et al., 2014; Stroeve et al., 1997). However, in mountainous regions satellite retrievals are complicated by relief, which impacts the at-sensor radiance measurements. Previous studies (Proy et al., 1989; Sandmeier and Itten, 1997) have shown that 1) the angle between the sun and the normal to the surface affects the irradiance received at the surface, 2) shadowed areas receive exclusively diffuse irradiance, 3) surrounding topography shields a part of the radiation from the sky, reducing the sky diffuse irradiance reaching the surface, but in turn 4) surrounding slopes reflect radiation to the surface, with potential multiple scattering with the atmosphere. For modelling purposes, the radiative fluxes contributing to the top-of-atmosphere (TOA) radiance over a mountainous scene can be broken down into different terms (Lenot et al., 2009), where the downwelling fluxes are split into four terms: direct, diffuse, reflections from neighbouring slopes, and the surface-atmosphere coupling. The upwelling fluxes are divided into the direct radiance, diffuse radiance and atmosphere intrinsic radiance. Most snow properties retrieval algorithms do not account for all of these terms, as they assume either a flat terrain, or only take into consideration the first order effect (tilt of the surface in the pixel) (e.g. Negi and Kokhanovsky, 2011; Stroeve et al., 2006; Teillet et al., 1982). However, neglecting the more complex terms of radiation budget in rugged terrain has been shown to introduce large uncertainties in satellite-based snow products (Dozier, 1980, 1989; Masson et al., 2018; Sirguey et al., 2009).

Efforts to account for the effects of rugged terrain on optical satellite measurements are numerous (e.g. Colby, 1991; Dozier, 1989; Duguay and Ledrew, 1992; Holben and Justice, 1980; Leprieur et al., 1988; Proy et al., 1989; Teillet et al., 1982; Woodham and Gray, 1987; Yang and Vidal, 1990). Early topographic normalisation methods were either limited by a loss of spectral information when relying on band-ratios (Holben and Justice, 1981), scene-dependent and poorly accurate when using image classification techniques (Conese et al., 1993b), or not suited for regions with pronounced topographic features when based on basic trigonometric corrections using a digital elevation model (DEM; Civco, 1989; Mishra et al., 2010) because surrounding relief effects were neglected, resulting in over-corrections (Conese et al., 1993a). The missing physical base in these methods led to the development of models simulating the solar radiation across the scene coupled with a radiative transfer model (e.g.



Sjoberg and Horn, 1983; Dozier and Frew, 1990; Dubayah and Rich, 1995; Richter, 1997; Sandmeier and Itten, 1997; Li et al.,
1999; Xin et al., 2002; Mousivand et al., 2015) to sum the contributions of the different terms to the measured TOA radiance.
Highly accurate 3D ray-tracing models have also been implemented to render satellite scenes (Gastellu-Etchegorry et al., 2004;
Poglio et al., 2006; Mayer et al., 2010), but their application is limited owing to the computational constraints of satellite image
processing.

Amongst the aforementioned physical models over rugged terrain, two different approaches are usually employed. The first
solves the inverse problem to obtain terrain-corrected surface reflectance maps from TOA images, whereas the second solves
the forward problem to generate TOA scenes as observed by a space-borne optical sensor from information about the surface
properties. Amongst the inverse models, the widely used ATCOR3 model (Richter, 1998) has been shown to perform well over
a variety of land surfaces (Dorren et al., 2003). More recently, Sen2Cor (Main-Knorn et al., 2017), based on ATCOR3 and
tailored for the Sentinel-2 satellites, and the multi-sensor MAJA (Lonjou et al., 2016) algorithm present in the ground segment
of CNES (THEIA, http://theia.cnes.fr, last access: May 12, 2020) were developed to produce atmosphere- and terrain-corrected
bottom-of-atmosphere (BOA) reflectance maps. These algorithms, which account for the first order reflections between slopes,
do not consider the multiple reflections of solar radiation between the terrain and the atmosphere. Yet, studies have shown that
neglecting the multiple reflections in snow-covered rugged areas leads to the underestimation of the irradiance received by the
surface, and hence the overestimation of ground reflectance (Dumont et al., 2011; Sirguey, 2009). Furthermore, because of
the strong anisotropy of snow (Warren, 1982), its bidirectional reflectance distribution function (BRDF) cannot be neglected
when using narrow field-of-view satellite sensors, particularly in rugged terrain where slopes introduce large variations in
illumination and viewing angles, and thus in the BRDF response. By default, ATCOR3 proposes an *a posteriori* empirical
BRDF correction scheme, albeit limited to faintly illuminated surfaces (Richter, 1998) and thus not adapted to snow-covered
scenes. The BREFCOR (Schlapfer et al., 2015) extension to ATCOR3 applies a BRDF correction following the iterative
atmospheric and topographic corrections, by tuning the Ross-Thick/Li-Sparse BRDF model (Schaaf et al., 2002) with a BOA
reflectance index. However, a subsequent study (Schlapfer et al., 2015) underlined the limitations of the model for snow
surfaces.

To the authors' knowledge, ModimLab (Sirguey et al., 2009) is the only inverse processing-chain specifically designed for
snow-covered surfaces able to account for the full effects of rugged terrain. MODImLab was developed to produce snow and
ice albedo, fractional cover, or Specific Surface Area (SSA) maps based on TOA radiance bands from the MODerate resolution
Imaging Spectroradiometer (MODIS). It uses approximations of the terms described by Richter (1998) for the definition of
the atmospheric and topographic effects, with the added benefit of using a reflectance model tailored for snow but based on
measurements (Dumont et al., 2011). The algorithm was later used to retrieve the surface albedo of glaciers in mountainous
regions (Brun et al., 2015; Dumont et al., 2012a, b; Sirguey et al., 2016), with an accuracy as good as $\pm 10\%$ (Dumont et al.,
2012b), a three-fold improvement over the commonly used MOD10 daily albedo product (Klein and Stroeve, 2002). The
sensor-dependence of MODimLab is limiting nevertheless, as it does not allow to exploit the wealth of new satellites currently
available.





Forward models of solar radiation over rugged terrain are scarcer than image correction algorithms. The SIERRA model (Lenot et al., 2009) allows the calculation of at-sensor radiance and can be inverted to perform the atmospheric and topo-

graphic corrections. Compared to the previously described models, SIERRA offers the advantages of accounting for multiple scattering and anisotropic surfaces using Rahman's parametric BRDF model (Rahman et al., 1993) despite the need for larger computational resources. However, the BRDF model poorly describes snow surfaces (Maignan et al., 2004), and although SIERRA achieves an excellent accuracy of 5% for reflectance retrievals over mountainous terrain (Lenot et al., 2009), the model was not tested over snow-covered surfaces nor for large solar zenith angles that occur during the winter. It is to be noted

that ATCOR4 (Richter and Schläpfer, 2002), designed for wide field-of-view optical airborne sensors, also provides a forward model to simulate radiance scenes from airborne images, but the applications are beyond the scope of this study.

The aforementioned examples highlight the lack of models tailored for retrieving snow properties from optical space-borne measurements, and thus a poor understanding of the effects of relief on these observations. The main motivations of this paper are to 1) assess the impact of the different topographic and atmospheric effects occurring in snow-covered mountainous re-

gions on TOA radiance, and 2 ) provide an assessment of errors when neglecting the full effects of rugged terrain by comparing simulations over rugged terrain with simulations that only consider flat terrain or slopes. To this end, the Radiative transfEr in ruggeD teRrain for rEmote Sensing of Snow (REDRESS) model was developed specifically for snow-covered surfaces, providing a modular tool that is easily applicable to different optical satellite sensors accounting for terrain/atmosphere coupling as well as the first order reflections, and using the asymptotic approximation of radiative transfer (AART) to describe the BRDF

of snow (Kokhanovsky and Zege, 2004; Kokhanovsky and Breon, 2012). In this study, REDRESS is described and evaluated by being run as a forward method to simulate TOA radiance scenes over the French Alps which are compared to Sentinel-3A OLCI scenes using *a priori* information about the physical properties of the snowpack and atmospheric conditions, a DEM of the scene under consideration, and the satellite sensor configuration as inputs.

Section 2 describes for the first time the detailed formulation of REDRESS and the approximations made for the different

fluxes considered, for which the theory is based on the forward formulation of Lenot et al. (2009), and the approximations of Sirguey et al. (2009). In Section 3, an overview of the model architecture is given, the datasets used for the validation of the model are presented, and the methods used to evaluated the model are detailed. The results of the simulations run for a test site in the French Alps are shown in Section 4, and the limitations of the study are discussed in Section 5.

## 2 Theory

Section 2.1 provides the first description of the formulation used in REDRESS to simulate TOA radiance over rugged topography, starting with the general formulation (Section 2.1.1), then detailing the solar fluxes reflected by the pixel observed by a satellite sensor (Section 2.1.2) and the contributions of the neighbouring slopes coupled with the atmosphere (Section 2.1.3). Second, Section 2.2 presents the simplified formulations of TOA radiance over flat terrain (Section 2.2.1) and considering slopes only (Section 2.2.2) used for the evaluation of the model.



## 2.1 TOA radiance formulation over rugged terrain

### 2.1.1 General formulation

The forward model computing TOA radiance over snow-covered rugged terrain is based on formulations and approximations found in Lenot et al. (2009), Sirguey et al. (2009), and Kokhanovsky and Zege (2004). To derive the equations involving the numerous terms contributing to TOA radiance, a simple and rigorous notation system is introduced in Table 1 and Figure 1. For example, following the notation system, $E_{hP}^{\text{flat}}$ represents the diffuse (h) downwelling irradiance (E) reaching a flat pixel under consideration ($P^{\text{flat}}$). Note that the wavelength $\lambda$ is omitted from all terms for clarity.

According to Lenot et al. (2009), the total radiance of a pixel P observed by a space-borne sensor can be expressed as:

$$L_{\text{TOA}}(P,\theta_v,\phi_v,\theta_i,\phi_i) = L_{tP}(P,\theta_v,\phi_v,\theta_i,\phi_i) + L_{t\mathcal{N}A}(P,\theta_v,\phi_v,\theta_i,\phi_i) + L_{tA}(\theta_v,\phi_v,\theta_i,\phi_i), \quad (1)$$

where $L_{tP}(P,\theta_v,\phi_v,\theta_i,\phi_i)$ is the solar radiation reflected by the pixel P directly towards the satellite's instantaneous field-of-view (IFOV), $L_{t\mathcal{N}A}(P,\theta_v,\phi_v,\theta_i,\phi_i)$ is the contribution of neighboring slopes (within a set distance to pixel P) scattered by the atmosphere to the satellite, and $L_{tA}(\theta_v,\phi_v,\theta_i,\phi_i)$ is the atmospheric path radiance measured by a sensor without any interaction with the surface. $\theta_i$, $\theta_v$, $\phi_i$, and $\phi_v$ describe the illumination and viewing zenith and azimuth angles respectively.

### 2.1.2 Contribution of the pixel's signal

The contribution to the TOA radiance of solar radiation reflected by the pixel P can be written as:

$$L_{tP}(P,\theta_v,\phi_v,\theta_i,\phi_i) = L_{dP}(P,\theta_v,\phi_v,\theta_i,\phi_i) + L_{hP}(P,\theta_v,\phi_v,\theta_i,\phi_i), \quad (2)$$

with the direct contribution of the reflected solar beam:

$$L_{dP}(P,\theta_v,\phi_v,\theta_i,\phi_i) = \Phi(P,\theta_v,\phi_v)\frac{\rho(P,\tilde{\theta}_i,\tilde{\theta}_v,\tilde{\phi})}{\pi}E_{dP}(P,\tilde{\theta}_i,\tilde{\phi}_i)T_{dir}\uparrow(P,\theta_v,\phi_v), \quad (3)$$

where $\rho(P,\tilde{\theta}_i,\tilde{\theta}_v,\tilde{\phi})$ is the bidirectional reflectance factor (BRF; Schaepman-Strub et al., 2006) of the surface (here $\tilde{\phi} = \tilde{\phi}_i - \tilde{\phi}_v$) and $T_{dir}\uparrow(P,\theta_v,\phi_v)$ is the direct atmospheric transmittance in the viewing direction, depending on the geographic location and elevation of the pixel P. The angles $\tilde{\theta}_{i,v}$ and $\tilde{\phi}_{i,v}$ are the effective (incident and viewing respectively) zenith and azimuth angles on a sloping surface at P, and are given by $\cos\tilde{\theta}_{i,v} = \cos\theta_{i,v}\cos\theta_n + \sin\theta_{i,v}\sin\theta_n\cos(\phi_{i,v}-\phi_n)$, with $\theta_n,\phi_n$ the slope and aspect of the surface respectively (Dumont et al., 2017). If the sensor has direct line of sight with the pixel P, the binary function $\Phi(P,\theta_v,\phi_v)$ is 1 and 0 otherwise. $E_{dP}(P,\tilde{\theta}_i,\tilde{\phi}_i)$ is the direct solar irradiance received by the surface, expressed as:

$$E_{dP}(P,\tilde{\theta}_i,\tilde{\phi}_i) = b(P,\theta_i,\phi_i)E_o\cos\tilde{\theta}_iT_{dir}\downarrow(P,\theta_i,\phi_i), \quad (4)$$





with $b(P,\theta_i,\phi_i)$, a sub-pixel fraction of shadows (Sirguey et al., 2009), where 0 corresponds to a fully shadowed pixel and 1 to a pixel free of shadows, accounting for self-shadows described by $\cos\tilde\theta_i <= 0$, and cast shadows calculated using a DEM (Dozier et al., 1981). $E_o$ is the extra-terrestrial solar irradiance and $T_{dir}\downarrow(P,\theta_i,\phi_i)$ is the direct atmospheric transmittance in the direction of the sun. As is the case for $T_{dir}\uparrow(P,\theta_v,\phi_v)$, $T_{dir}\downarrow(P,\theta_i,\phi_i)$ depends on the location and altitude of the pixel P.

The diffuse contribution of the reflected solar radiation by pixel P is expressed as:

$$L_{hP}(P,\theta_v,\phi_v,\theta_i,\phi_i) = \Phi(P,\theta_v,\phi_v)\frac{a_v(P,\tilde\theta_v)}{\pi}E_{hP}(P,\tilde\theta_i,\tilde\phi_i)T_{dir}\uparrow(P,\theta_v,\phi_v), \tag{5}$$

where $a_v$ is the hemispheric-directional reflectance (HDR, Schaepman-Strub et al., 2006) of the pixel that reads:

$$a_v(P,\theta_v) = \frac{1}{\pi}\int\limits_{\theta=0,\phi=0}^{\pi/2,2\pi}\!\!\!\!\rho(P,\theta,\theta_v,\phi)\cos\theta d\Omega, \tag{6}$$

with $d\Omega = \sin\theta d\theta d\phi$. $E_{hP}(P,\tilde\theta_i,\tilde\phi_i)$ is the diffuse irradiance from the surrounding slopes and the atmosphere received by pixel P, and written:

$$E_{hP}(P,\tilde\theta_i,\tilde\phi_i) = E_{hP}^{\text{flat}}(P,\theta_i,\phi_i)V_d(P) + E_{tGP}(P,\tilde\theta_i,\tilde\phi_i) + E_{tGAP}(P,\tilde\theta_i,\tilde\phi_i), \tag{7}$$

with $E_{hP}^{\text{flat}}(P,\theta_i,\phi_i)$ the irradiance received by a theoretically horizontal surface modulated by $V_d(P)$, the sky-view factor (Dozier et al., 1981) varying from 0 to 1, and expressed by Sirguey et al. (2009) as:

$$V_d(P) = \frac{1}{2\pi}\int\limits_0^{2\pi}\big(1-\cos H(\phi)\big)d\phi, \tag{8}$$

for which the horizon elevation $H(\phi)$ for a given azimuth $\phi$ is calculated using Dozier et al. (1981)'s algorithm. $E_{tGP}(P,\tilde\theta_i,\tilde\phi_i)$ is the irradiance received from surrounding slopes, and $E_{tGAP}(P,\tilde\theta_i,\tilde\phi_i)$ is the coupled atmospheric irradiance reaching pixel P after multiple reflections by the surrounding slopes and scattering by the atmosphere. When taking into account these adjacency effects, the diffuse radiance of the pixel P (Equation 5) at the $k^{\text{th}}$ iteration becomes:

$$L_{hP}^{(k)}(P,\theta_v,\phi_v,\theta_i,\phi_i) = \Phi(\tilde\theta_v,\tilde\phi_v)\frac{a_v(P,\tilde\theta_v,\tilde\phi_v)}{\pi}T_{dir}\uparrow(P,\theta_v,\phi_v)$$
$$\Big[E_{hP}^{\text{flat}}(P,\theta_i,\phi_i)V_d(P) + E_{tGP}^{(k)}(P,\tilde\theta_i,\tilde\phi_i) + E_{tGAP}^{(k)}(P,\theta_i,\phi_i)\Big], \tag{9}$$





and is updated at each iteration $k$. According to Sirguey et al. (2009), the solar radiation reflected once by a neighboring slope towards the pixel under consideration, can be approximated as:

$$E_{tGP}^{(k)}(P,\tilde{\theta}_i,\tilde{\phi}_i) = E_{tP}^{\text{flat}}(P,\theta_i,\phi_i)\frac{\left(1 - V_d(P)\right)\iint_{\mathcal{N}(P)}R^{(k-1)}(M)dS_M}{1 - \iint_{\mathcal{N}(P)}R^{(k-1)}(M)dS_M\iint_{\mathcal{N}(P)}\left(1 - V_d(M)\right)dS_M}. \tag{10}$$

The total downwelling irradiance $E_{tP}^{\text{flat}}(P,\theta_i,\phi_i) = E_{dP}^{\text{flat}}(P,\theta_i,\phi_i) + E_{hP}^{\text{flat}}(P,\theta_i,\phi_i)$ is expressed for a horizontal surface,

where the diffuse solar irradiance $E_{hP}^{\text{flat}}(P,\theta_i,\phi_i)$ is calculated with an atmospheric radiative transfer model, and the direct solar irradiance is expressed as:

$$E_{dP}^{\text{flat}}(P,\theta_i,\phi_i) = E_o\cos\theta_i T_{dir}\downarrow(P,\theta_i,\phi_i). \tag{11}$$

The bi-hemispherical albedo of the surface of a pixel M $S(M)$, located in $\mathcal{N}(P)$, is expressed as:

$$S(M) = \frac{1}{\pi}\int\limits_{\theta=0,\phi=0}^{\pi/2,2\pi}\!\!\!\!\!a_s(M,\theta,\phi)\cos\theta d\Omega, \tag{12}$$

where $a_s$ is the surface directional-hemispherical reflectance (DHR; Schaepman-Strub et al., 2006). Similarly to Equation 6, the DHR of a pixel M can be written:

$$a_s(M,\theta_i) = \frac{1}{\pi}\int\limits_{\theta=0,\phi=0}^{\pi/2,2\pi}\!\!\!\!\!\rho(P,\theta_i,\theta,\phi)\cos\theta d\Omega, \tag{13}$$

and thus for any point X, $a_s(X,\theta_i) = a_v(X,\theta_i)$. $R$ is the surface hemispherical-conical reflectance (HCRF) (Schaepman-Strub et al., 2006) of the pixel P, is updated with the values calculated for the previous iteration $k-1$, and written:

$$R(P,\theta_v,\phi_v,\theta_i,\phi_i) = \pi\frac{\rho(P,\tilde{\theta}_v,\tilde{\theta}_i,\tilde{\phi})E_{dP}(P,\tilde{\theta}_i,\tilde{\phi}_i) + a_v(P,\tilde{\theta}_v,\tilde{\phi}_v)E_{hP}(P,\tilde{\theta}_i,\tilde{\phi}_i)}{E_{dP}(P,\tilde{\theta}_i,\tilde{\phi}_i) + E_{hP}(P,\tilde{\theta}_i,\tilde{\phi}_i)}$$

$$= \pi\frac{L_{hP}(P,\theta_i,\theta_v,\phi_i,\phi_v) + L_{dP}(P,\theta_i,\theta_v,\phi_i,\phi_v)}{T_{dir}\uparrow(P,\theta_v,\phi_v)\Phi(\tilde{\theta}_v,\tilde{\phi}_v)\left(E_{dP}(P,\tilde{\theta}_i,\tilde{\phi}_i) + E_{hP}(P,\tilde{\theta}_i,\tilde{\phi}_i)\right)}$$

$$= \pi\frac{L_{TOA}(P,\theta_i,\theta_v,\phi_i,\phi_v) - L_{t\mathcal{N}A}(P,\theta_i,\theta_v,\phi_i,\phi_v) - L_{tA}(\theta_i,\theta_v,\phi)}{T_{dir}\uparrow(P,\theta_v,\phi_v)\Phi(\tilde{\theta}_v,\tilde{\phi}_v)\left(E_{dP}(P,\tilde{\theta}_i,\tilde{\phi}_i) + E_{hP}(P,\tilde{\theta}_i,\tilde{\phi}_i)\right)}. \tag{14}$$





In mountainous areas, when the trapping mechanism of reflected radiation between slopes is considered (Sirguey, 2009), Equation 10 is modified to account for the coupling:

$$E_{tGP}^{(k)}(P, \tilde{\theta}_i, \tilde{\phi}_i) = \left( E_{tP}^{\text{flat}}(P, \theta_i, \phi_i) + E_{tGAP}^{(k)}(P, \theta_i, \phi_i) \right)$$
$$\frac{\left(1 - V_d(P)\right) \iint_{\mathcal{N}(P)} R^{(k-1)}(M) dS_M}{1 - \iint_{\mathcal{N}(P)} R^{(k-1)}(M) dS_M \iint_{\mathcal{N}(P)} \left(1 - V_d(M)\right) dS_M}, \tag{15}$$

assuming that all the pixels in the neighbourhood $\mathcal{N}$ are receiving the same irradiance as a horizontal surface, i.e. $E_{tP}^{\text{flat}}(M, \theta_i, \phi_i)$ and that the surface reflectance of the surrounding pixels is lambertian, thus $R(M, \theta_i^M, \theta_v^M, \phi_i^M, \phi_v^M) = S(M)$. In the literature, the neighbourhood of reflected radiation by surrounding terrain $\mathcal{N}(P)$ is generally delimited by a circle of 0.5–1 km (Lenot et al., 2009; Richter, 1998; Sirguey et al., 2009). The HCRF of the pixels in the neighborhood, $R^{(k-1)}(M) = R^{(k-1)}(M, \theta_i^M, \theta_v^M, \phi_i^M, \phi_v^M)$ is calculated with Equation 14.

The atmospheric coupling contributing to pixel P's illumination, which accounts for the multiple reflections between slopes (also present in Equation 15) can also be approximated iteratively according to Sirguey et al. (2009):

$$E_{tGAP}^{(k)}(P, \theta_i, \phi_i) = E_{tP}^{\text{flat}}(M, \theta_i, \phi_i) \frac{\alpha_{\text{atm}} \iint_{\mathcal{N}_2(P)} R^{(k-1)}(M) dS_M}{1 - \alpha_{\text{atm}} \iint_{\mathcal{N}_2(P)} R^{(k-1)}(M) dS_M}, \tag{16}$$

with $\alpha_{\text{atm}}$, the atmospheric hemispherical albedo, which is the diffuse descending atmospheric component. Based on previous studies, the neighbourhood used for the contribution of pixels to the diffuse atmospheric scattering ($\mathcal{N}_2(P)$) differs from the neighbourhood used for the effects of scattering by surrounding slopes ($\mathcal{N}(P)$), and is in the range 1–2 km (Lenot et al., 2009; Richter, 1998; Sirguey et al., 2009).

### 2.1.3 Contributions of surrounding slopes

Following Equation 1, the contribution of the slopes in the vicinity of the pixel under consideration to the measured TOA radiance (without any interaction with the pixel P but nevertheless a dependency to P to define the neighborhood) can be divided into three components (Figure 1):

$$L_{t\mathcal{N}A}(P, \theta_v, \phi_v, \theta_i, \phi_i) = L_{tGA}(P, \theta_v, \phi_v, \theta_i, \phi_i) + L_{tGGA}(P, \theta_v, \phi_v, \theta_i, \phi_i) + L_{tGAGA}(P, \theta_v, \phi_v, \theta_i, \phi_i), \tag{17}$$

where $L_{tGA}(P, \theta_v, \phi_v, \theta_i, \phi_i)$ corresponds to the solar radiation scattered once by the surrounding ground, then by the atmosphere towards the sensor, $L_{tGGA}(P, \theta_v, \phi_v, \theta_i, \phi_i)$ is the contribution of the multiple reflections occurring between slopes, and $L_{tGAGA}(P, \theta_v, \phi_v, \theta_i, \phi_i)$ is the coupled atmospheric multiple scattering. It follows that:

$$L_{tGA}(P, \theta_v, \phi_v, \theta_i, \phi_i) = \iint_{\mathcal{N}(P)} \left[ \frac{a_s\left(M, \tilde{\theta}_i(M), \tilde{\phi}_i(M)\right)}{\pi} E_{dP}\left(M, \tilde{\theta}_i(M), \tilde{\phi}_i(M)\right) F_{env}(M, \theta_v, \phi_v) \right] dS_M, \tag{18}$$





where $\tilde{\theta}_i(M)$ and $\tilde{\phi}_i(M)$ denote the effective solar zenith and azimuth angles accounting for the local topography of pixel M. $F_{env}$ is the weight of the pixel M in the TOA radiance originating from $\mathcal{N}(P)$. $F_{env}$ accounts for both terrain and atmospheric effects (i.e. atmospheric transmittance) and can be calculated using a Monte-Carlo approach (Lenot et al., 2009). However, the approximation of Equation 17 used in this study (see Equation 20) does not require such calculations. The two remaining terms of Equation 17 are considered to be intertwined and hence are defined as:

$$210 \quad L_{tGGA}(P,\theta_v,\phi_v,\theta_i,\phi_i) + L_{tGAGA}(P,\theta_v,\phi_v,\theta_i,\phi_i) = \iint_{\mathcal{N}(P)} \frac{S_M}{\pi} E_{hP}(M,\tilde{\theta_i}^M,\tilde{\phi_i}^M) \quad F_{env}(M,\theta_v,\phi_v)dS_M.$$ (19)

Although the different terms of Equation 17 have be introduced separately above, Sirguey et al. (2009) have shown that the total contributions of surrounding slopes to TOA radiance can be approximated overall as:

$$L_{t\mathcal{N}A}^{(k)}(P,\theta_v,\phi_v,\theta_i,\phi_i) = \frac{t_d(\theta_v,\phi_v)}{\pi}\overline{R}_e^{(k-1)}(P,\theta_v,\phi_v,\theta_i,\phi_i)\Big( E_{dP}^{\text{flat}}(P,\theta_i,\phi_i) + E_{hP}^{\text{flat}}(P,\theta_i,\phi_i)$$
$$+ E_{tGAP}^{(k)}(P,\theta_i,\phi_i)\Big),$$ (20)

where $t_d(\theta_v,\phi_v)$ is the atmospheric diffuse transmittance, expressed by Sirguey et al. (2009) as:

$$215 \quad t_d(\theta_v,\pi_v) = \frac{E_{hP}^{\text{flat}}(P,\theta_v,\phi_v)}{E_o\cos\theta_v},$$ (21)

which corresponds to the radiation scattered by the atmosphere reaching the sensor, and can be calculated using an atmospheric radiative transfer model by imposing the sensor geometry $(\theta_v,\phi_v)$ in place of the sun angles. The environmental reflectance defined by Sirguey et al. (2009) as the spatial average of each pixel reflectance $R$ within the neighbourhood $\mathcal{N}_2$ is iteratively updated:

$$220 \quad \overline{R}_e^{(k-1)} = \frac{\iint_{\mathcal{N}_2(P)} R^{(k-1)}(M)dS_M}{\iint_{\mathcal{N}_2(P)} dS_M}.$$ (22)

## 2.2 Simplified cases of TOA radiance formulation

To provide a comparison of REDRESS output with the current approaches generally used in the literature and assess the errors introduced by neglecting the rugged terrain effects, REDRESS was run considering a horizontal surface or slopes only, by updating the terms of the general formulation (Equation 1) for each configuration.





### 2.2.1 Flat terrain

For a perfectly flat landscape, the TOA radiance is limited to the sum of the direct and diffuse downwelling irradiance reflected by the pixel's surface and the atmospheric path radiance. In this configuration, the signal measured by a space-borne sensor is expressed as:

$$L_{TOA}(P,\theta_v,\phi_v,\theta_i,\phi_i) = L_{dP}^{\text{flat}}(P,\theta_v,\phi_v,\theta_i,\phi_i) + L_{hP}^{\text{flat}}(P,\theta_v,\phi_v,\theta_i,\phi_i) + L_{tA}(P,\theta_v,\phi_v,\theta_i,\phi_i), \tag{23}$$

where the contribution of the reflected direct radiance is written:

$$L_{dP}^{\text{flat}}(P,\theta_v,\phi_v,\theta_i,\phi_i) = \frac{\rho(P,\theta_i,\theta_v,\phi)}{\pi} E_{dP}^{\text{flat}}(P,\theta_i,\phi_i) T_{dir} \uparrow (P,\theta_v,\phi_v), \tag{24}$$

and the reflected diffuse solar radiation originating from the scattered irradiance by the atmosphere only is:

$$L_{hP}^{\text{flat}}(P,\theta_v,\phi_v,\theta_i,\phi_i) = \frac{a_v(P,\theta_v,\phi_v)}{\pi} E_{hP}^{\text{flat}}(P,\theta_i,\phi_i) T_{dir} \uparrow (P,\theta_v,\phi_v). \tag{25}$$

In this study, the simulations performed with REDRESS using the flat terrain configuration are referred to as REDRESS$_{\text{flat}}$.

### 2.2.2 Sloping terrain

When considering the effects of terrain slope and aspect, the flat terrain formulation is modified to account for the local illumination and viewing angles modified by the terrain, shadowing, and the surfaces visible by the satellite sensor. Therefore, in the formulation developed in Section 2.2.1, Equation 24 is replaced by Equation 3, and Equation 25 is replaced by Equation 5 in which the reflected illumination and atmospheric / terrain coupling are neglected. Thus, in Equation 5 the diffuse contribution of the downwelling solar radiation (Equation 7) is reduced to:

$$E_{hP}(P,\theta_v,\phi_v,\theta_i,\phi_i) = E_{hP}^{\text{flat}}(P,\theta_i,\phi_i) V_d(P). \tag{26}$$

The simulations performed with REDRESS, accounting for the first order slope effects only, are noted REDRESS$_{\text{slope}}$ hereinafter.

## 3 Data and Methods

The section first presents the implementation of the model, followed by the data and methods used to validate the model.



### 3.1 Model implementation

The equations in Section 2 form a forward model aiming to solve Equation 1, and thus generate synthetic TOA spectral radiance images over snow-covered mountainous terrain as measured by an optical space-borne sensor. The model was implemented in Python with an architecture organised around interchangeable modules (Figure 2, pink boxes) designed to be easily replaced, allowing the application of different formulations at each different computation stage. The processing workflow consists of five modules that read and process the input data (green in Figure 2), providing the model's configuration parameters, and one core module (beige box in Figure 2) that performs the iterative forward calculations of TOA radiance. The four main data sources required to run the model and the modules used to process the inputs, as well as the main module are described in the following sections.

#### 3.1.1 Satellite sensor configuration

Information about the satellite's sensor configuration is used as inputs to the forward model, enabling a direct comparison between the model outputs and TOA radiance measured by a specified satellite sensor. The data extracted from the satellite product include $\theta_i, \phi_i$ and $\theta_v, \phi_v$, the illumination and viewing angles at the time of the overpass, the spatial resolution, and the spectral characteristics of the sensor. For each spectral channel, the radiance is computed with a 1 nm wavelength step and integrated using the spectral response function of the satellite bands. In this study, the module used to read the satellite data is based on the Python API of the ESA Sentinel Application Platform (SNAP, http://step.esa.int, last access: May 12, 2020), but other reading modules can be easily implemented.

#### 3.1.2 Snow physical properties

Specific surface area (SSA) is one of the main drivers of snow reflectance. It is a metric directly related to the optical diameter of snow grains which is widely used in remote sensing of snow (e.g. Dozier, 1984; Scambos et al., 2007; Painter et al., 2009). The current version of the model relies on the assumption that the SSA is known and constant across the scene. The SSA is used as an input of the asymptotic analytical radiative transfer theory, to compute surface Directional-Hemispherical Reflectance (DHR, a.k.a. black-sky albedo) and BiHemispherical Reflectance (BHR, a.k.a. white-sky albedo) and the BRF of the snow surface (Kokhanovsky and Zege, 2004; Kokhanovsky and Breon, 2012). The resulting BRF is used to compute the direct contribution of the reflected solar beam for each pixel, as shown in Equation 3.

#### 3.1.3 Atmospheric properties

The atmospheric components are calculated using a transfer model that is initialised with four main parameters: water vapour content, the total ozone column, the type of aerosol present in the atmosphere and the total aerosol optical depth (AOD) obtained from the datasets described in Section 3.2.4. Additionally, 6S takes as inputs $\theta_i, \phi_i$ and $\theta_v, \phi_v$, the solar and viewing angles extracted from the satellite product and the ground elevation obtained from the DEM. The model provides the required atmospheric inputs of REDRESS, which include: $E_o$, the extra-terrestrial solar irradiance, $E_{hP}^{\text{flat}}$, the diffuse downwelling



irradiance at the surface, $T^{\uparrow}_{dir}$ and $T^{\downarrow}_{dir}$, the atmospheric transmittance in the solar and viewing directions, $a_s$, the atmosphere's spherical albedo, $L_{tA}$, the atmosphere's intrinsic radiance, and the direct-to-diffuse illumination ratio at the surface. In the current setup, REDRESS, written in python, uses the Py6S module (Wilson, 2013) to run the 6S (Vermote et al., 1997) Fortran

code, but the model is designed for an easy implementation of other radiative transfer codes.

### 3.1.4  Topographic parameters

A DEM is used to compute all the topographic parameters as follows. The slope and aspect of the terrain across the scene $(\theta_n, \phi_n)$ are computed using Horn (1981)'s third order finite difference weighted by the reciprocal of distance. The algorithm, which has been shown to represent slope and aspect more precisely than other approaches (Lee and Clarke, 2005), does not

overestimate the slopes or exaggerate peaks, although a loss in local variability is observed (smoothing of small dips and peaks). The fraction of sky masked by surrounding relief for a given point is described by the sky-view factor (Dubayah and Rich, 1995), which is calculated using Dozier et al. (1981)'s DEM-based horizon algorithm. As described in further detail by Sirguey et al. (2009), the horizon algorithm is run for each cell of the DEM in 64 azimuth directions to compute the horizon elevation angle, which in turn is used to determine the sky-view factor (Equation 8). The horizon algorithm is also used to compute shad-

ows, represented by a binary map where 0 represents a shadowed pixel, and 1 a sunlit pixel. The total shadow product is the combination of self-shadows (i.e. slopes that do not face the sun) and cast-shadows (i.e. areas where solar radiation is blocked by surrounding features). Self-shadows are usually found where the cosine of the effective solar zenith angle is negative. However, visual comparisons with high-resolution imagery showed that using a slightly positive cut-off value (0.035) to account for the inaccuracies in the DEM allows a better representation of self-shadows. Cast-shadows are defined here as the areas where

the horizon elevation angle in the direction of the sun is larger than the solar zenith angle. Noise and isolated pixels are removed by applying a 3×3 binary dilatation-erosion step (using the Python functions scipy.ndimage.morphology.grey_dilation and scipy.ndimage.morphology.grey_erosion) to the cast-shadows map. All the topographic products are first calculated at the DEM resolution, then are resampled to the satellite resolution and in the geometry of acquisition of each image. The resampling steps are performed in the code using the GDAL library (GDAL/OGR contributors, 2019).

### 3.1.5  Iterative forward module

Using the inputs described in the previous sections, the TOA radiance of each pixel across the scene is calculated iteratively. To start, the BHR of the snow is used as a first guess in the iteration process for the diffuse environmental reflectance and the radiation reflected by the adjacent slopes. The terms are updated at each iteration from the calculated HCRF (Equation 14) averaged with a sliding window of the size of the neighbourhood contributing to the pixel (Equations 15 and 16). In turn, the

radiance of the neighboring terrain (Equation 17), the diffuse contribution of the reflected solar radiation (Equation 5), and the total radiance (Equation 1) are updated. Numerous calculations over different datasets have shown that the iteration process convergence is rapid, with on average 4–6 iterations over a typical alpine scene with pronounced topography. The iterative process is terminated when the average difference in radiance across the scene is less than 0.1% between two consecutive iterations.





### 3.2 Study site and data

#### 3.2.1 Study site

A study area covering approximately 14 km × 18 km in the southwestern French Alps was selected for the evaluation of the model (Figure 3). The region is composed of narrow valleys surrounded by steep slopes, with elevations ranging from 1520 m to 3983 m. The extent includes Emparis (45°04′N, 6°14′E), a large grass plateau with moderate relief located at an elevation of approximately 2000 m, and Col du Lautaret (45°02′N, 6°24′E), a high-altitude mountain pass that reaches an elevation of 2058 m, characterised by a wide open area above the tree line surrounded by high mountain peaks. To highlight specific terrain configurations and examine the spectral performance of the model, four individual sites were selected across the scene (shown in Figure 3). The first point, *P1*, is located on the Emparis plateau, and is the closest configuration to flat terrain with slightly sloping terrain and very little contribution from nearby features. *P2* is located on a small plateau at the Col du Lautaret site, where field measurements were acquired (see Section 3.2.3). The pixel *P2* presents small slopes with nevertheless an important contribution of the surrounding topography. The third site, *P3*, located to the south of the Col du Lautaret pass, is characterised by a steep north-facing slope and is fully shadowed. Lastly, *P4* is located on a sunlit south-facing slope located to the north of the Col du Lautaret where the direct solar radiation dominates the signal. A summary of the four locations and their characteristics is presented in Table 2.

#### 3.2.2 Satellite data

The OLCI (Ocean and Land Colour Instrument) sensor on-board Sentinel 3 is a push-broom imaging spectrometer that covers the spectral range 400–1020 nm with 21 bands and has a spatial resolution of approximately 300 m at nadir. The instrument provides a high repeat coverage, with a revisit time inferior to 2 days in the French Alps, and a large spatial coverage thanks to a swath width of 1270 km, providing an ideal tool for monitoring the dynamics of alpine snow. The TOA of atmosphere radiance simulations for the aforementioned study area were compared to five Sentinel-3A OLCI L1 full-resolution scenes acquired between 13 February 2018 and 06 April 2018. Overpass dates were selected based on 2 criteria: the absence of clouds, which was visually assessed on-site, and the acquisition of concurrent surface SSA measurements in the field. The five dates matching the criteria cover a wide-range of snow conditions from winter to spring, and were acquired with different solar and viewing geometries, allowing the validation of the model for a wide range of acquisition conditions. For the detailed evaluation purposes in Section 4.1, the scene acquired on 13 February 2018 was retained, as it contained the most snow-covered pixels. A summary of the satellite acquisitions is presented in Table 3.

#### 3.2.3 Field measurements

Ground measurements were collected during the Sentinel-3 overpass for the five dates. The SSA of snow samples taken at the surface of the snowpack was measured across the Col du Lautaret site using the Alpine Snowpack Specific Surface Area Profiler (ASSSAP) (Arnaud et al., 2011), which has an estimated accuracy of 10 %. The measurements taken on 13 and 21





February, 14 March, and 6 April were carried out on a small plateau located on the south-facing slopes to the north of the site, crossing pixel P2, whereas on 22 March SSA measurements were acquired across the north-facing slopes located on the south side of Col du Lautaret pass, approximately halfway between P2 and P3. For both sites, the measurements were taken along transects covering a variety of slopes and orientations, and thus snow conditions. SSA samples were measured every 20 m along the transects. The regular spacing between the points was ensured by using a 20 m rope, and the start and end points of each transect were geolocalised using a hand-held GPS with an estimated accuracy of ±5m. The result of the measurements are given in Table 3.

### 3.2.4 Model setup

For each Sentinel-3 OLCI acquisition, TOA radiance was simulated for the 21 spectral bands using REDRESS. The model was run using the ancillary data from the Sentinel-3 acquisition as input parameters: solar and viewing zenith and azimuth angles, sensor characteristics, and the spectral band response functions. The average SSA measured along the transect was used as input SSA, given that the model considers a fixed SSA value across the scene. The topography was obtained from the Advanced Spaceborne Thermal Emission and Reflection Radiometer (ASTER) Global Digital Elevation Model (GDEM) Version 3 (NASA/METI/AIST/Japan Spacesystems, 2019), with a spatial resolution of 1 arc second (approximately 30 m). The DEM was selected for its suitable resolution for the computation of topographic parameters in mountainous regions (Frey and Paul, 2012) and its wide-spread availability. The input parameters for the radiative transfer calculations performed with Py6S were obtained from different sources. The Copernicus Atmosphere Monitoring Service (CAMS) near-real time analysis dataset provides daily analyses of atmospheric conditions and aerosol content with a horizontal resolution of 0.4° (approximately 31x44 km). The total column ozone, total column water vapour, and total aerosol optical depth (AOD) at 550 nm were extracted from this dataset for each OLCI product. For all the simulations, the aerosol model was set to the standard 6S "Continental" model (Vermote et al., 2006). The radiative transfer model was run twice for the entire study area: a first time with the average viewing and illumination geometries of the scene, and a second time inverting the viewing and illumination geometries to compute the missing parameters (e.g. Equation 21). Finally, the iterative model was run with the environmental reflectance neighborhood $\mathcal{N}_2$ (see Equation 16) set to 2.1 km, and the neighborhood of the radiation reflected by surrounding terrain, $\mathcal{N}$ (see Equation 15) set to 1.5 km, which were found to yield the best results for the selected scene.

The current implementation of the model only considers snow-covered pixels, and does not account for fractional-snow cover. Therefore, for validation purposes, the TOA radiance was not modelled for pixels containing less than 80% of snow. The snow cover percentage was obtained by calculating the proportion of Theia Snow collection pixels (Gascoin et al., 2019), based on Sentinel-2 images at a resolution of 20 m, within each Sentinel-3 OLCI pixel. Due to the lower revisit time of Sentinel-2 (5 days), the closest available date to the OLCI image was used, and cloud covered areas were filled with the next closest date.

### 3.3 Surface reflectance

Surface reflectance retrieved from TOA radiance satellite observations is a common product of interest which is more widely used in Earth Observation than TOA radiance itself. Although the inversion of surface reflectance was not performed in RE-





DRESS, a first estimation of the impact of the complex topographic effects on future retrievals is proposed by converting
the TOA radiance simulations to bottom-of-atmosphere (BOA) HCRF (hereinafter referred to as reflectance) by removing
the atmospheric terms. This conversion was performed for the scene on 13 February 2018, first for REDRESS simulations,
then considering the effect of slopes only. The surface reflectance was obtained from TOA radiance using Equation 14, which
removes the atmospheric terms. For the slope only configuration, Equation 14 was modified to remove the contribution of
neighbouring terrain (Equation 17) and the reflected solar radiation from neighbouring slopes (Equation 10), thus becoming:

$$R(P, \theta_v, \phi_v, \theta_i, \phi_i) = \frac{L_{\text{TOA}}(P, \theta_i, \theta_v, \phi_i, \phi_v) - L_{tA}(\theta_i, \theta_v, \phi)}{T_{dir}^\uparrow(P, \theta_v, \phi_v)\Phi(\tilde{\theta}_v, \tilde{\phi}_v)} \frac{\pi}{\left( E_{dP}(P, \tilde{\theta}_i, \tilde{\phi}_i) + E_{hP}(P, \tilde{\theta}_i, \tilde{\phi}_i) \right)} \quad (27)$$


### 3.4 Sensitivity analysis

To evaluate the sensitivity of the TOA simulations to the model input parameters, the Rugged Terrain model was run for
varying values of snow SSA, AOD, total column water vapour, and total ozone column, reported in Table 4. The simulations
were performed with the configuration of 13 February 2018. During the simulations over a range of values for each input
parameter, the other inputs were fixed to the values reported in Table 3. The range of values for snow SSA was taken from the
minimum and maximum values measured in the field over the snow season 2017–2018, and for the atmospheric parameters,
from the minimum and maximum values in the CAMS near-real time analysis dataset between December 2017 and April 2018.

## 4 Results

The performance of REDRESS is first evaluated by comparing in detail the model output to the Sentinel-3 OLCI TOA radiance
image acquired on the 13 February 2018 (Section 4.1), and second by extending the comparison to five Sentinel-3 OLCI
acquisitions over an entire winter season (Section 4.2). Finally, the sensitivity of the model results to the input parameters is
investigated (Section 4.3).

### 4.1 Model performance on a single date

#### 4.1.1 Spatial performance

Figure 4 shows a Sentinel-3 OLCI L1B image subset covering the study area (red box in Figure 3) acquired on 13 February
2018, compared to a TOA radiance synthetic map produced with REDRESS at 510 nm (band 05) and 1020 nm (band 21).
The observations and simulations highlight that the TOA signal is dominated by the complexity of the topography and that
the range of observed TOA radiance values is indeed high for seemingly uniform snow-covered surfaces. Overall, an excellent
agreement between the measured and modelled TOA radiance is observed at both wavelengths, highlighting the model's ability
to reproduce the large variations in TOA radiance across the scene despite the same snow intrinsic properties being applied to
all pixels. Indeed, the spatial pattern between the satellite and synthetic images are similar, with dark shaded areas located in





north-facing slopes and brighter south-facing slopes (at the time of the acquisition, the sun was located south-southeast, see Table 3). Nevertheless, the variations in TOA radiance are more pronounced in the synthetic image, which has a slightly larger value range (45–511 $\mathrm{W\,m^{-2}\,sr^{-1}\,\mu m^{-1}}$ at 510 nm; 2–181 $\mathrm{W\,m^{-2}\,sr^{-1}\,\mu m^{-1}}$ at 1020 nm) than that of the satellite image

(36–456 $\mathrm{W\,m^{-2}\,sr^{-1}\,\mu m^{-1}}$ at 510 nm; 2–162 $\mathrm{W\,m^{-2}\,sr^{-1}\,\mu m^{-1}}$ at 1020 nm), and cause the scene to appear sharper. At both wavelengths, the TOA radiance of south-facing slopes (i.e. in the southern part of the images) is weakly over-estimated by the model, making them appear brighter than in the satellite image, whereas shadows located on north-facing slopes are less gradual than in the satellite image and form denser dark areas where the TOA radiance is under-estimated.

To illustrate the performance of the model, the scatterplots and bias histograms between the measured and synthetic images

are shown for the study area at 510 nm (Figure 5a and b) and 1020 nm (Figure 6a and b). At both wavelengths, the values are densely distributed along the identity line, with a correlation of 0.71 at 510 nm and 0.73 at 1020 nm. The bias is low (11 $\mathrm{W\,m^{-2}\,sr^{-1}\,\mu m^{-1}}$ at 510 nm and 6.7 $\mathrm{W\,m^{-2}\,sr^{-1}\,\mu m^{-1}}$ at 1020 nm), and the histogram of the model errors reveals a peaked, nearly centred distribution with a long tail skewed toward the positive values (over-estimation of the model), as well as a smaller tail covering negative values. To identify the shortcomings of the model, the values over- and under-estimated by

more than 2 standard deviations of the bias were colored in red and blue respectively, and identified as such in all the panels of Figures 5 and 6. An enlarged subset of the overlay of the corresponding pixels is shown on a high-resolution panchromatic SPOT 6 image acquired on 19 February 2019, 6 days after the OLCI scene and at a similar time of day (Figures 5c and 6c). At both wavelengths, the pixels over-estimated by the model are distributed along ridgelines or buttresses, where a change in the slope and aspect is observed, and the pixels under-estimated by the model are located on the edge of projected shadowed

areas. Removing the red and blue areas from the analysis significantly improves the correlation, yielding $r^2$=0.84 at 510 nm and $r^2$=0.86 at 1020 nm. However, in this study all the pixels were kept for the analysis.

### 4.1.2   Spectral performance

The comparison between the TOA radiance spectra measured by Sentinel-3 OLCI and simulated using REDRESS on 13 February 2018 is shown in Figure 7 for four pixels located across the study site (Figure 3). As was the case for the two individual

wavelengths in Figure 4, a good agreement between the REDRESS simulated spectra and the TOA radiance measured by Sentinel-3 OLCI is observed at all wavelengths, despite the different terrain configuration of the four pixels (Table 2). To assess the errors caused by neglecting the full effects of rugged terrain, simulations considering a flat surface (REDRESS$_{\mathrm{flat}}$) and slopes (REDRESS$_{\mathrm{slope}}$) are represented by dashed and dotted lines respectively. In the case of the pixel P1 (Figure 7a), which is the closest to a flat surface, the modelled TOA radiance with REDRESS$_{\mathrm{flat}}$ is, as expected, overall close to the satellite

measurement, with a mean absolute error (MAE) of 9.8 $\mathrm{W\,m^{-2}\,sr^{-1}\,\mu m^{-1}}$. Modelling the TOA radiance with REDRESS$_{\mathrm{slope}}$ slightly improves the results compared to flat terrain above 510 nm, but underestimates the signal at shorter wavelengths leading to an increased MAE of 16.2 $\mathrm{W\,m^{-2}\,sr^{-1}\,\mu m^{-1}}$. Even in such a configuration with small slopes (<10°) and negligible re-illumination from neighbouring slopes, running REDRESS largely improves the simulation results, with a MAE of 3.1 $\mathrm{W\,m^{-2}\,sr^{-1}\,\mu m^{-1}}$. For terrain configurations with slightly larger slopes and pronounced surrounding features such as P2 (Fig-

ure 7b), accounting for the slopes brings improvement over considering a flat terrain (MAE of 21.1 and 38.1 $\mathrm{W\,m^{-2}\,sr^{-1}\,\mu m^{-1}}$





respectively) across all spectral bands, but similarly to the other sites, underestimates the measured TOA signal. In comparison, REDRESS simulations perform well, with a MAE of 4.8 $\mathrm{W\,m^{-2}\,sr^{-1}\,\mu m^{-1}}$. In more extreme cases, the flat terrain assumption leads on the one hand to a significant over-estimation of the TOA radiance in steep north-facing shadowed slopes (P3, Figure 7c) mainly because it does not consider shadowing due to relief, resulting in a MAE of 102.8 $\mathrm{W\,m^{-2}\,sr^{-1}\,\mu m^{-1}}$. On the other

hand, the TOA radiance of strongly re-illuminated south-facing slopes (P4, Figure 7d) is under-estimated by REDRESS$_{\mathrm{flat}}$, with a MAE of 126.1 $\mathrm{W\,m^{-2}\,sr^{-1}\,\mu m^{-1}}$. Accounting for slopes shows a better agreement with the satellite measurements, however the lack of consideration of multiple reflections between the pixel and surrounding slopes, as well as the atmospheric and terrain coupling lead to a systematic under-estimation of the measured TOA signal (MAE of 23.4 $\mathrm{W\,m^{-2}\,sr^{-1}\,\mu m^{-1}}$ for P3 and 30.7 $\mathrm{W\,m^{-2}\,sr^{-1}\,\mu m^{-1}}$ for P4). REDRESS is able to simulate the large variations from a pixel to another, satisfac-

torily reproducing the satellite measurements, with a MAE of 1.7 $\mathrm{W\,m^{-2}\,sr^{-1}\,\mu m^{-1}}$ for P3 and 7.3 $\mathrm{W\,m^{-2}\,sr^{-1}\,\mu m^{-1}}$ for P4. Despite a good overall agreement, small discrepancies are observed in the range 510–620 nm, which may be explained by uncertainties in the parameterisation of the atmospheric ozone content in the radiative transfer module, as highlighted in Section 4.3.

### 4.1.3    Contributing terms to TOA radiance in snow-covered rugged terrain

The discretisation of the different terms making up the TOA radiance simulated with the full Rugged Terrain model underlines the spectrally-dependent contribution of the terms to the overall signal. For the sites P1, P2, and P4, located in direct sunlight, the reflected direct sunlight (L$_{\mathrm{dP}}$) is of the first order, dominating the signal in the near-infrared domain (between 90 and 95% at 1020 nm), and making up almost half the signal in the blue range of the spectrum (between 44 and 56% at 400 nm). Furthermore, the slope has a strong influence on the intensity and spectral shape of L$_{\mathrm{dP}}$, in turn largely impacting the overall

spectral shape of the total TOA radiance (Picard et al., 2020). P3 differs strongly from the other sites owing to its position on a self-shadowed slope, leading to L$_{\mathrm{dP}}$ being null. All the other components of the reflected solar radiation are diffuse, and therefore contribute decreasingly to the total signal as a function of wavelength between the blue and near-infrared. The reflected sky radiation (L$_{\mathrm{dP}}$) depends on the proportion of sky "seen" by the pixel, expressed by $V_d$ (Equation 8, shown in Figure 7). Indeed, the contribution of L$_{\mathrm{hP}}$ is more important in an open area such as P1, with $V_d$=0.98 (41 $\mathrm{W\,m^{-2}\,sr^{-1}\,\mu m^{-1}}$ at 400

nm) than for P3 with $V_d$=0.77 (34 $\mathrm{W\,m^{-2}\,sr^{-1}\,\mu m^{-1}}$ at 400 nm) which is located in the middle of a steep slope and thus with part of the sky masked by its own slope as well as by surrounding slopes. The contributions of the multiple reflections from surrounding slopes and the coupled terrain–atmosphere reflections (L$_{\mathrm{tGP}}$ and L$_{\mathrm{tGAP}}$) largely depend on the terrain surrounding the observed pixel. Hence, the contributions of surrounding slopes are small for the flat plateau on which P1 is located, with a maximum of 5 $\mathrm{W\,m^{-2}\,sr^{-1}\,\mu m^{-1}}$ at 490 nm, whereas for P4, surrounded by large slopes, the contribution reaches 33

$\mathrm{W\,m^{-2}\,sr^{-1}\,\mu m^{-1}}$. For the four sites presented in Figure 7, L$_{\mathrm{tGP}}$ makes up most of the contribution from the surrounding terrain, whereas L$_{\mathrm{tGAP}}$ is negligible. The neighboring pixels' diffuse and coupled atmospheric contribution (L$_{\mathrm{t}\mathcal{N}\mathrm{A}}$) is similar for all sites, varying between 31 $\mathrm{W\,m^{-2}\,sr^{-1}\,\mu m^{-1}}$ (P1) and 33 $\mathrm{W\,m^{-2}\,sr^{-1}\,\mu m^{-1}}$ (P4) at 400 nm. Lastly, for all sites, the contribution of the atmosphere intrinsic radiance is commensurate for all pixels, as it depends on the illumination and viewing geometries, the altitude of the pixel and the atmospheric composition, which are all similar for the four sites.



Over snow-covered surfaces, all the terms making up the TOA radiance are of the same order of magnitude, except the terrain-atmosphere coupling (multiple reflections between slopes) which is negligible. Therefore to correctly infer information about the surface from space-borne images, all of the following terms need to be considered: i) the slope and aspect of the surface, ii) re-illumination from neighbouring slopes, and iii) atmospheric scattering into the sensor's field of view. Neglecting the two last terms leads to significant under-estimations of the TOA radiance.

## 4.2   Model performance over a winter season

To assess the performance of REDRESS over a snow season, the model was applied to the 5 Sentinel-3 OLCI acquisitions between 13 February and 06 April 2018. The results are presented in the form of the distribution of the difference between the model outputs and Sentinel-3 OLCI images at 510 nm and 1020 nm in Figure 8. In order to gauge the benefits of using the REDRESS model over more common approaches considering the first-order effects of slopes only, the figure also shows the

bias distribution of simulations performed with REDRESS$_{slope}$. Similarly to Figure 4, pixels containing less than 80% snow were discarded from the analysis. Thus, the number of pixels used for the comparison has a decreasing trend throughout the season owing to snow-melt (Table 3).

### 4.2.1   REDRESS simulations

The bias distribution between REDRESS simulations and Sentinel-3 OLCI images shown in Figure 8 is similar for the five

dates in 2018, demonstrating the model's consistency in simulating TOA radiance for a variety of snow cover and properties conditions. Thus, it is reasonable to consider the performance analysis in Section 4.1 to be representative for the simulations performed throughout the winter season. At 510 nm, the bias remains low for four first dates, ranging from 5 $\mathrm{W\,m^{-2}\,sr^{-1}\,\mu m^{-1}}$ on 21 February 2018 to -17 $\mathrm{W\,m^{-2}\,sr^{-1}\,\mu m^{-1}}$ on 14 March 2018. The slightly more pronounced over-estimation of REDRESS observed on 6 April 2018 (22 $\mathrm{W\,m^{-2}\,sr^{-1}\,\mu m^{-1}}$) could be attributed either to a mis-representation of the snow-cover in the

Theia Snow collection product used to mask snow-free pixels, or to the presence of impurities in the snow. The RMSE values are consistent between the dates ranging 55–72 $\mathrm{W\,m^{-2}\,sr^{-1}\,\mu m^{-1}}$. No seasonal trends in the simulation errors are evident at 510 nm (band 05), which is similar to that of the range 400–865 nm (data not shown here).

Conversely at 1020 nm, the bias between REDRESS and the Sentinel-3 OLCI images is low (<7 $\mathrm{W\,m^{-2}\,sr^{-1}\,\mu m^{-1}}$) between 13 February and 14 March 2018, then increases towards the end of the snow season as the error distribution shifts towards an

over-estimation of the TOA radiance. For the two last dates, a slight over-estimation of the model is observed, with bias values of 8 and 28 $\mathrm{W\,m^{-2}\,sr^{-1}\,\mu m^{-1}}$ on 22 March and 6 April 2018 respectively. Furthermore, a widening of the distribution of the errors manifesting itself as smaller centered peaks in the histograms and a steady increase in RMSE (from 26 $\mathrm{W\,m^{-2}\,sr^{-1}\,\mu m^{-1}}$ on 13 February 2018 to 36 $\mathrm{W\,m^{-2}\,sr^{-1}\,\mu m^{-1}}$ on 6 April 2018), and that does not occur at shorter wavelengths, is observed throughout the season. This trend may be explained by the increase in SSA variability that generally occurs in the Spring

(Bühler et al., 2015), and that is not accounted for in the model. Indeed, spatially heterogeneous SSA values may cause the measured SSA at the Col du Lautaret site to be unrepresentative of the snow conditions present across the study site, potentially playing a role in the over-estimation of the TOA radiance in the near-infrared.



### 4.2.2 Comparison with the first order slope approach

At 510 nm, the distribution of the errors REDRESS$_{slope}$ has a similar pattern to that of REDRESS, and no significant difference

in correlation is observed from one model configuration to the other. However, only considering the slopes leads to a systematic under-estimation of the TOA radiance, which may be explained by the lack of consideration of additional solar radiation received by the pixel coming from surrounding slopes and scattered by the atmosphere. This under-estimation becomes larger as the season advances, most likely owing to an increasingly higher solar position resulting in increased multiple reflections between slopes, which were correctly estimated by REDRESS. Improvements in bias going from REDRESS$_{slope}$ to REDRESS

simulations range between 20 and 81 $\mathrm{W\,m^{-2}\,sr^{-1}\,\mu m^{-1}}$, reducing the bias values to the range -17–22 $\mathrm{W\,m^{-2}\,sr^{-1}\,\mu m^{-1}}$. These improvements in bias are of the same order in the range 400–865 nm (not shown here). Furthermore, the RMSE is increasingly improved between model configurations throughout the season in link with the trend in bias observed, with a decrease of 4 $\mathrm{W\,m^{-2}\,sr^{-1}\,\mu m^{-1}}$ on 13 February 2018 and 46 $\mathrm{W\,m^{-2}\,sr^{-1}\,\mu m^{-1}}$ on 06 April 2018, leading to values of 66 and 55 $\mathrm{W\,m^{-2}\,sr^{-1}\,\mu m^{-1}}$ for the respective dates.

As was the case at 510 nm, the error distribution between REDRESS and REDRESS$_{slope}$ at 1020 nm is similar, with a slightly negative bias obtained with REDRESS$_{slope}$ for all dates except the last. Both model configurations perform closely for the first three dates, with bias values of -1, -7, and -9 $\mathrm{W\,m^{-2}\,sr^{-1}\,\mu m^{-1}}$ observed with REDRESS$_{slope}$ versus 7, 4, and 3 $\mathrm{W\,m^{-2}\,sr^{-1}\,\mu m^{-1}}$ with REDRESS, which improves the results slightly on 21 February and 14 March 2018. The improvements in bias are less pronounced at 1020 nm than at shorter wavelengths, which may be explained by the smaller proportion of

diffuse solar radiation in the infrared. The RMSE values remain similar (<2 $\mathrm{W\,m^{-2}\,sr^{-1}\,\mu m^{-1}}$) for the two configurations. For the two last dates, REDRESS over-estimates the TOA radiance by a larger amount than REDRESS$_{slope}$. The overestimation is particularly marked on 6 April 2018, where the bias of REDRESS$_{slope}$ (11 $\mathrm{W\,m^{-2}\,sr^{-1}\,\mu m^{-1}}$) is more than halved compared to REGRESS, and the difference in RMSE between the model configurations is 10 $\mathrm{W\,m^{-2}\,sr^{-1}\,\mu m^{-1}}$. However, if the assumption of a unrepresentative SSA value at the end of the season holds true, the systematic under-estimation of TOA radiance observed

on the other dates with REDRESS$_{slope}$ would be compensated by this effect, thus deceivingly reducing the bias.

### 4.3 Sensitivity to input parameters

The impact of the model's input parameters on the simulated TOA radiance is shown in Figure 9 for the four pixels located across the study site (Table 2) on 13 February 2018. The Sentinel-3 OLCI TOA radiance (colored squares) is compared to the simulations based on the values indicated in Table 3 (dashed lines) and considering a range for each parameter (Table

4) (shaded areas). The variations in SSA between 3 and 88.5 $\mathrm{m^2\,kg^{-1}}$, which correspond to a range of snow conditions from melting spring snow (6 April 2018) to freshly deposited wind-blown snow (13 February 2018), have an increasingly large influence on the TOA radiance between 500 and 1020 nm, with a maximal effect at 1020 nm as expected (Warren and Wiscombe, 1980). The influence of SSA on TOA radiance is closely linked to the terrain configuration. At 1020 nm, the TOA radiance varies by 91 $\mathrm{W\,m^{-2}\,sr^{-1}\,\mu m^{-1}}$ across the SSA range for P4, 66 $\mathrm{W\,m^{-2}\,sr^{-1}\,\mu m^{-1}}$ for P2, 45$\mathrm{W\,m^{-2}\,sr^{-1}\,\mu m^{-1}}$ for

P1, and 5 $\mathrm{W\,m^{-2}\,sr^{-1}\,\mu m^{-1}}$ for P3. The percentage difference between the TOA radiance calculated with the minimum and



maximum SSA values is similar for pixels P1 (74%) and P2 (78%), and P4 (80%) but slightly higher for the steep shadowed pixel P3 (97%). Over the wavelength range 865–1020 nm, the modelled TOA radiance using the average SSA value measured in the field fit well the measured TOA for all sites, with a small bias of 17 $\mathrm{W\,m^{-2}\,sr^{-1}\,\mu m^{-1}}$ for P2, which is unexpected, as the SSA was measured at that location. This discrepancy may be explained by the fact that the SSA measured along the transect on 13 February 2018 was sampled from a heterogeneous snowpack at the meter-scale, spanning a wide range of values, from 21.9 to 88.5 $\mathrm{m^2 kg^{-1}}$ (Table 4), and therefore the average SSA value (41.5 $\mathrm{m^2 kg^{-1}}$) was not representative of the pixel. At P3 this SSA range causes a 21 $\mathrm{W\,m^{-2}\,sr^{-1}\,\mu m^{-1}}$ variation in TOA radiance at 1020 nm (data not shown here), underlining the challenges of characterising surface snow properties at a 300 m scale.

Contrarily to SSA, variations in Aerosol Optical Depth (AOD) have a large impact on the TOA radiance throughout the wavelength range (Figure 9b). The TOA radiance decreases between simulations with an AOD of 0.0075 and 0.2169 for the sites P1, P2, and P4, which are located in direct sunlight, with a stronger effect on pixels with larger slopes (P4 > P2 > P1). Indeed, the TOA radiance decreases by a factor of approximately 1.3 when going from the minimum to the maximum AOD for P4, 1.05 for P1, 1.15 for P2. This increase is of the same order at all wavelengths. In contrast the shadowed pixel P3 benefits from increased scattering in the atmosphere, and thus the TOA radiance increases by a factor 1.5 at 400 nm to 3.5 at 1020 nm when going from an AOD of 0.0075 to 0.2169. Furthermore, at P3 the change is an increasing function of the wavelength. The results suggest that AOD was correctly predicted by the CAMS product on 13 February 2018, as the simulated TOA radiance using the reanalysis value of 0.019 closely matches the measured spectra. Nevertheless, the large change introduced by varying the AOD highlights the importance of correctly parameterising or retrieving the AOD in the atmospheric radiative transfer model.

Ozone and water vapour in the atmosphere (Figure 9c and d respectively) have an effect on a limited number of bands only. Variations in ozone affect the TOA signal between 490 (band 4) and 681.25 nm (band 10). As observed for SSA, the relative variations of TOA radiance in regards to the ozone total column in the atmosphere are similar for the four studied sites. For pixel P3, variations are negligible, with a difference of 3 $\mathrm{W\,m^{-2}\,sr^{-1}\,\mu m^{-1}}$ at 560 nm, whereas for pixel P4, the change in TOA radiance reaches 23 $\mathrm{W\,m^{-2}\,sr^{-1}\,\mu m^{-1}}$. At 560 and 620 nm, where the effect of ozone is the strongest (Chappuis, 1880), REDRESS systematically over-estimates the TOA radiance at all four sites when using a value of 0.0084 $\mathrm{kg\,m^{-2}}$ (395 DU). However, using the range of forecasted values over the winter season, between 0.006 and 0.010 $\mathrm{kg\,m^{-2}}$ (281–477 DU) is not sufficient to explain the TOA radiance at the two bands, as the satellite measurements lie outside the range of simulated TOA radiance for the two sites with larger TOA radiance values (P2 and P4). Given that the bands perturbed by water vapour (bands 18, 885 nm and 20, 940 nm) were *a priori* removed for the analysis in this study, the only wavelength affected by water vapour is 900 nm (band 19). A large increase in the total column water vapour from 0.79 to 13.81 $\mathrm{kg\,m^{-2}}$ causes a similar decrease of approximately a factor 1.2 across all sites. On 13 February 2018, the measured TOA radiance falls between the range of predicted values at 940 nm, and for all sites except P1 for which the model more generally over-estimates radiance in the near-infrared using the CAMS prediction of 1.75 $\mathrm{kg\,m^{-2}}$ leads to a good agreement between the modelled and measured TOA radiance.



## 5 Discussion

### 5.1 Implications for surface reflectance estimation

To infer surface geophysical properties from optical remote sensing data, it is common to first estimate surface reflectance by applying atmospheric corrections and second use the obtained surface reflectance spectra in subsequent retrieval algorithms. Here using REDRESS, the benefits of considering the full rugged terrain problem over simply accounting for slope effects on future BOA reflectance retrievals are evaluated. For this, the TOA radiance synthetic maps produced with REDRESS and REDRESS$_{slope}$ for the OLCI scene on 13 February 2018 were converted to BOA reflectance (Figure 10). The synthetic BOA reflectance images are shown at 510 and 1020 nm, covering the study area delimited in Figure 3. At both wavelengths, the images produced with REDRESS have relatively uniform values across the scene, with a mean value of 0.97 ±0.03 (1σ) at 510 nm and 0.77 ±0.03 (1σ) at 1020nm. The BOA reflectance values vary across the image according to the slope and aspect of the terrain, with the DEM spatial signal clearly appearing. The model is able to simulate the reflectance in the shadowed areas well, although the variability is partly lost and thus reflectance appears as uniform patches in the shade. The images produced with REDRESS$_{slope}$ are more spatially variable (mean values of 1.45 ±0.51 (1σ) and 1.39 ±1.35 (1σ) at 510 and 1020 nm respectively). Furthermore, the BOA reflectance produced with REDRESS$_{slope}$ is systematically higher than with REDRESS, with average values at all wavelengths across the map over 1. This over-estimation is exacerbated in the shadowed areas of the image, where reflectance values are almost always over 1.5 at 510 nm (orange shaded areas in Figure 10), and for a large proportion at 1020 nm.

Examples of the BOA reflectance spectra at all bands are shown in Figure 11 for the four pixels across the study site (Table 2). The spectra simulated using REDRESS (solid lines and squares) show a similar spectral shape, typical of fresh snow, for the four sites with values varying between 0.96 (P2) and 1.01 (P4) at 400 nm and 0.74 (P2) and 0.79 (P3) at 1020 nm. The small differences observed between the sites are explained by changes in illumination and viewing angles caused by the slope and aspect of the surface (Picard et al., 2020). On the other hand, the BOA reflectance obtained with REDRESS$_{slope}$ feature a marked decreasing trend in the 400–620 nm range for the sunlit pixels P1, P2, and P4 (decrease between 0.19 and 0.23), which differs from the flat response obtained with REDRESS (decrease between 0.01 and 0.03). In the near-infrared, the BOA reflectance obtained with REDRESS$_{slope}$ is closer to that of REDRESS with a difference between the simulation configurations of 0.04, 0.02, and 0.05 for pixels P1, P2 and P4 respectively. The spectrum simulated with REDRESS$_{slope}$ configuration is not shown for pixel P3, since it is shadowed, leading to highly unrealistic values which are often masked out or ignored (Scambos et al., 2007; Crawford et al., 2013). Figures 10 and 11 suggest that considering the full effects of rugged terrain (Equation 1) would noticeably improve the retrievals of snow physical properties in shadowed areas. Furthermore, although the simulated reflectance for sunlit slopes is similar in the infrared wavelengths whether the full rugged terrain effects or just the slopes are considered due to the small proportion of diffuse illumination at those wavelengths, Figure 11 heralds that considering the full rugged terrain problem will significantly improve reflectance retrievals in the visible wavelengths.





## 5.2   Limitations and further improvements

Although REDRESS shows promising results for the simulation and then correction of satellite scenes over snow-covered surfaces with complex topographic features, a number of uncertainties are introduced into the product along the process.

First, the accuracy of the topographic products depend on the DEM used as input. The artefacts found in terrain changes in the results (Figures 5 and 6) in relationship with shadows and ridges confirm the finding of Lenot et al. (2009), in which retrievals over rugged terrain were shown to be mostly impacted by the coarse resolution of the DEMs typically used or the errors in the orthorectification between the DEM and the satellite image (Richter, 1998). Here, the observed errors along the ridges may be due to the resolution of the DEM (30 m) or a mis-representation of the terrain, since the the overall vertical

accuracy has been estimated to be 10–25 m (ASTER GDEM Validation Team, 2009), which is significant for calculating topographic parameters. Furthermore, even in the case of an accurate DEM, the resolution of the DEM can introduce errors at the ridges due to a smoothing effect of the topography leading to an over-estimation of the radiance. Olson et al. (2019) showed that the incident shortwave radiation is overestimated in mountainous regions as the resolution of the DEM becomes coarser. Oppositely, the distribution of the pixels for which the TOA radiance is largely under-estimated by the model along

the borders of the shadowed areas in the image suggests that the calculation of the shadow extent is over-estimated in the model, with sunlit pixels falling into the shadows. The ASTER-GDEM was selected in this study for its near-global coverage and resolution ratio of 0.1 in comparison to the Sentinel-3 OLCI image, following Richter (1998) who recommend a DEM resolution of 0.25 times the satellite image size or better. The availability of high-resolution DEM products when applying the model to higher resolution imagery (e.g. Sentinel-2) is the main limiting factor of the method and further research is required

on how to accurately represent the terrain at different spatial scales.

Second, the results in Figure 9 also highlight how critical the atmospheric radiative transfer calculations are at all wavelengths, the modelled TOA radiance having been shown here to be sensitive to the parameterisation of the atmosphere in the model. Figure 9 puts in evidence the sensitivity of the TOA radiance to the AOD at 550 nm, even if the CAMS near-real-time product appears to have correctly evaluated the atmospheric parameters on the investigated dates. Nevertheless, small

errors in the estimation of the AOD at the time of the satellite overpass may result in non-trivial errors in the retrieval of snow surface properties, with an over-estimation of the AOD causing an over-estimation of the radiance for sunlit pixels and an under-estimation for shadowed pixels. The possibility of error is exacerbated by the fact that it is not uncommon to see factor 3–5 changes in AOD from a day to another in the Alps (Lenoble et al., 2008). Additionally, the atmospheric radiative transfer calculations consider clear skies only and do not account for cloud cover. Cloud masking algorithms have been shown

to perform badly over snow-covered terrain (Stillinger et al., 2019) and were chosen not to be applied to this work because the small extent of the studied area allowed visual selection of the images.

Third, in this study the TOA radiance is simulated with the same SSA across the scene, each scene having a different SSA, potentially introducing errors in the near-infrared due to spatial variations in the surface snow properties. For the scene modelled on 13 February 2018, the assumption of a uniform snow cover was reasonably valid as a snowfall occurred less

than 24h before the satellite overpass. Despite these favorable conditions a larger dispersion is obtained at 1020 nm (Figure 6)





compared to 510 nm (Figure 5) suggesting possible uncertainties owing to surface properties spatial variations. With the aim of using the model to retrieve surface parameters from satellite images, it will be possible to iteratively calculate and update the snow physical properties in every pixel. As well, as considering a uniform SSA, the current application of REDRESS assumes a 100% snow cover, and snow free pixels are removed *a posteriori* based on an external product. Therefore, errors in the snow-

cover product used to classify the pixels will result in large uncertainties in the model outputs, with an over-estimation of the TOA radiance for pixels erroneously detected as snow-covered. To overcome the limitation, and for the large-scale application of REDRESS, further developments to iteratively estimate fractional snow-cover derived from surface reflectance (Gascoin et al., 2019; Hall et al., 2002) are forseen.

Although the calculation of the terms making up the TOA radiance over rugged terrain in REDRESS is computationally

cheap, processing the topographic parameters used as inputs and running the Py6S radiative-transfer model considerably slow the process. The computational times were not of concern for the evaluation site selected in this study, nevertheless these two points are to be considered when running the model over larger areas such as an entire mountain range. First, the calculation of the horizons in 64 azimuth directions for each pixel (described in Section 3.1.4) is the most time-consuming process when running REDRESS, with an execution time of approximately 23 minutes for an image of 1000x1000 pixels on a standard

desktop computer (2 CPUs, 2.0 Ghz processor, 7.5 GB RAM). However, the model allows to choose between calculating the horizons products as needed or using pre-calculated products. Thus for larger extents, the horizon products can be pre-calculated using the dedicated tool, a step that only needs to be performed once. Second, the model setup used in this study only performs the radiative-transfer calculations twice (Section 3.2.4) using averaged input values for the entire study area. The extent of the study area was considered sufficiently small for the differences in illumination and viewing geometries (typically

<0.5°) as well as atmospheric parameters to be negligible across the scene. However, for large study areas the radiative-transfer model needs to be run on a pixel-per-pixel basis to account for the sometimes large variations in the model inputs observed at a regional scale. This approach is not feasible with the current setup which takes approximately 2 seconds per pixel to compute the atmospheric parameters. Given the modular nature of REDRESS, the use of a radiative-transfer tool dedicated to satellite imagery correction, or the adaptation of Py6S for a use with pre-calculated Lookup Tables are recommended.

**6   Conclusions**

A new modular approach designed to simulate TOA radiance measured by space-borne optical sensors over snow-covered rugged terrain is presented. REDRESS, comprised of a DEM-based forward model associated with a radiative transfer model for the atmosphere and a dedicated snow BRDF model, allows to estimate the different terms contributing to the measured signal, and is adapted to account for highly reflective surfaces, for which scattering effects are exacerbated.

The model was applied to five Sentinel-3 OLCI scenes acquired over an entire winter season in the French Alps. For each date, REDRESS was initialised with snow SSA measurements performed in the area at the time of the satellite overpass, and using the CAMS daily analysis of atmospheric conditions as input for the atmospheric radiative transfer module. Results show that REDRESS is able to simulate TOA radiance images of snow-covered rugged terrain comparable to those measured by a
multi-spectral optical space-borne sensor, as long as particular care is taken in the selection of the DEM and input source for the atmospheric parameters. Synthetic TOA images modelled using the full terrain problem show higher levels of agreement with the Sentinel-3 OLCI scenes compared to images simulated considering the effects of slopes only, particularly in shadowed areas or steep slopes. Furthermore, the study highlights the large variations of the terms making up the total TOA radiance in rugged terrain. Over the winter season (February – April 2018) the contribution to the total TOA signal of the radiation reflected from surrounding slopes towards observed sunlit pixels was in the range 6–10%, and the diffuse contribution of multiple reflections from the neighbourhood coupled with the atmosphere represented 0.5–7% of the signal, both with a larger contribution in the visible wavelengths. In the shadowed areas, the diffuse contribution of the neighbourhood was similar to that of the sunlit pixels. The contribution of surrounding slopes to the pixel's radiance was similar to the sunlit pixels in the visible wavelengths (6–8%) but significantly larger in the near-infrared, ranging from 8 to 40% across the season, owing to the other diffuse contributions being small. Because the reflected signal is weak in the near-infrared, the neighbourhood contributions make up a larger proportion of the signal. The analysis of the different terms contributing to the TOA radiance has shown that the contributions of surrounding terrain cannot be neglected in the visible wavelengths even for small slopes (<10°), and has an important effect in shadowed areas in the infrared wavelengths. Accounting for the full rugged terrain problem is thus essential for optical satellite remote sensing observations over snow-covered mountainous terrain.

*Code availability.* The source code of the REDRESS model, its documentation and examples will be made available on GitHub upon publication.

*Data availability.* Sentinel-3A OLCI scenes were sourced as open data from the European Space Agency (ESA) Copernicus Open Access Hub (https://scihub.copernicus.eu/dhus/). The SPOT6 image was obtained via the CNES Kalideos Alpes project (https://alpes.kalideos.fr). The atmospheric CAMS data was obtained via the ECMWF catalog (https://www.ecmwf.int/). The Sentinel-2 snow cover data was sourced from the Theia platform (https://www.theia-land.fr/product/neige/).

*Author contributions.* M. Lamare, M. Dumont and G. Picard designed the study and developed the theory. M. Lamare, C. Delcourt, F. Tuzet, F. Larue and L. Arnaud contributed to the acquisition of the field data. M. Lamare led the analysis and wrote the manuscript. All authors contributed to revisions of the manuscript.

*Competing interests.* The authors declare that they have no conflict of interest.



*Acknowledgements.* This study was funded by APR CNES MIOSOTIS, EBONI ANR-16-CE01-006 and BNP Paribas foundation. CNRM/CEN

and IGE are part of Labex OSUG@2020 (investissement d'avenir – ANR10 LABX56). This work has been carried out using the resources made available by the Copernicus Research User Support (RUS) service (http://rus-copernicus.eu), funded by the European Commission, managed by the European Space Agency, and operated by CS SI and its partners. The authors are grateful to Lautaret staff and Station Alpine Joseph Fourier (SAJF) for instrument maintenance and for supporting the *in situ* experiments.





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



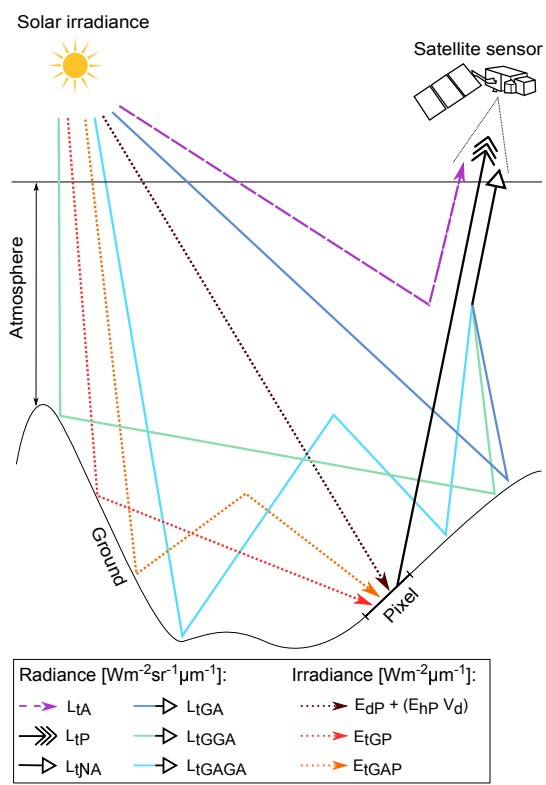

**Figure 1.** Illustration of the contributing fluxes to the TOA radiance measured by a space-borne sensor. The dotted lines correspond to solar irradiance received by the pixel P which are then reflected towards $L_{tP}$, the sensor's instantaneous field-of-view (IFOV), the solid colored lines represent the contribution of neighbouring slopes to the measured signal ($L_{t\mathcal{N}A}$), and the dashed line is the atmospheric path radiance ($L_{tA}$). For sake of clarity, the direct and diffuse components of the downwelling fluxes are merged in this scheme.





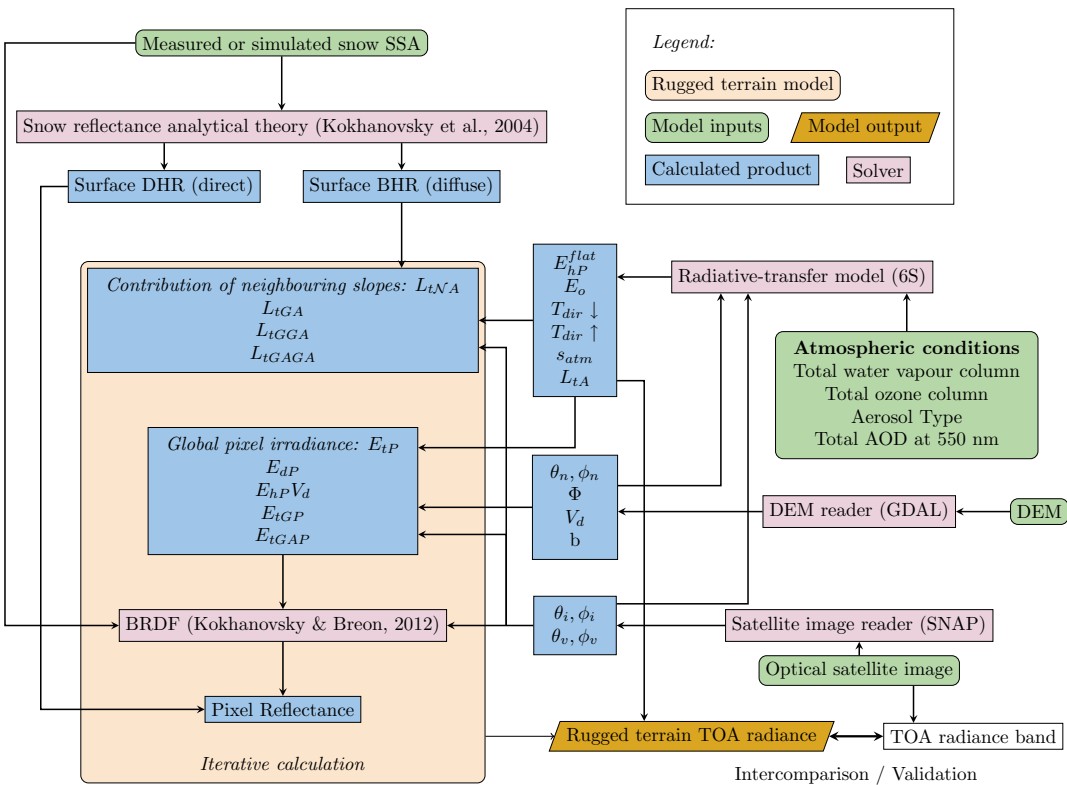

**Figure 2.** Processing steps to simulate TOA radiance over snow-covered rugged terrain. The model input products (green boxes) and the variables (blue boxes) are used in the iterative process (beige box) that calculates TOA radiance (orange lozenge). The pink boxes represent the interchangeable modules in the model, with the formulation used in this study in brackets.



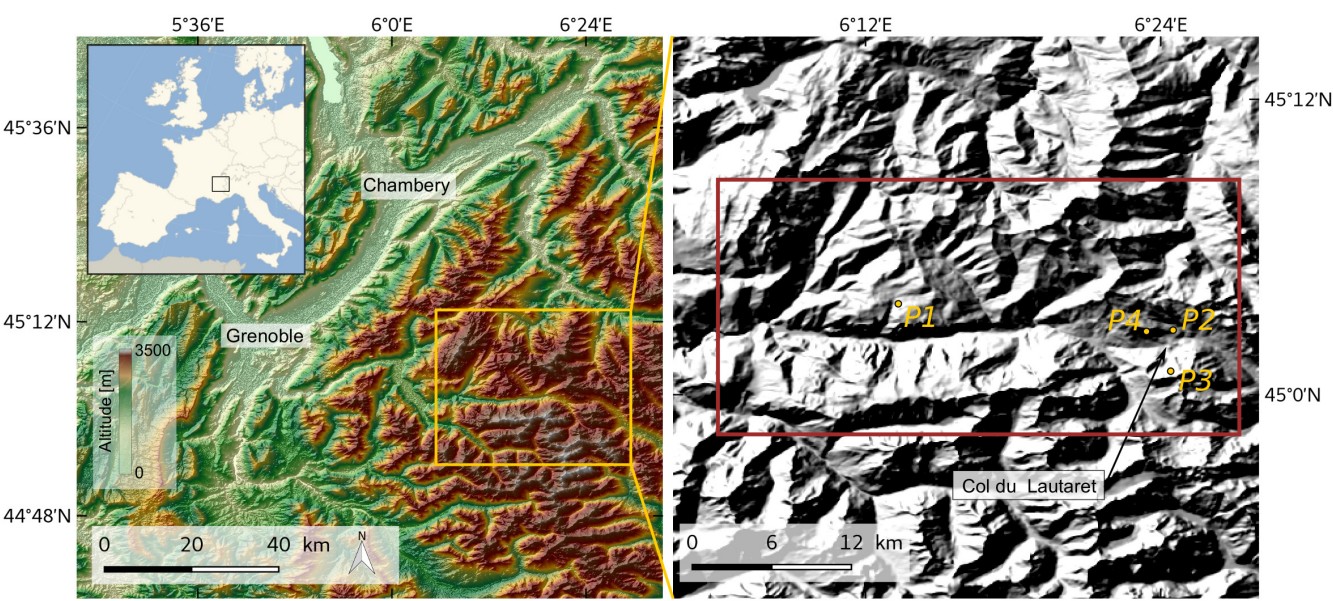

**Figure 3.** Location of the Col du Lautaret experimental site. The extent used for the validation of the model is outlined in red, over the hillshade product generated from and draped over the 30m ASTER DEM. The four individual sites are marked as yellow dots. (Europe map source: Alexrk2 / Wikimedia Commons, modified under the Creative Commons Attribution-Share Alike 3.0 Unported license).



**Figure 4.** TOA radiance image observed by Sentinel-3 OLCI L1B (© Copernicus) acquired on 13 February 2018 (left) and simulated with the Rugged Terrain model (right) at 510 nm (top) and 1020 nm (bottom). Pixels containing less than 80% of snow were masked out (hatched region).



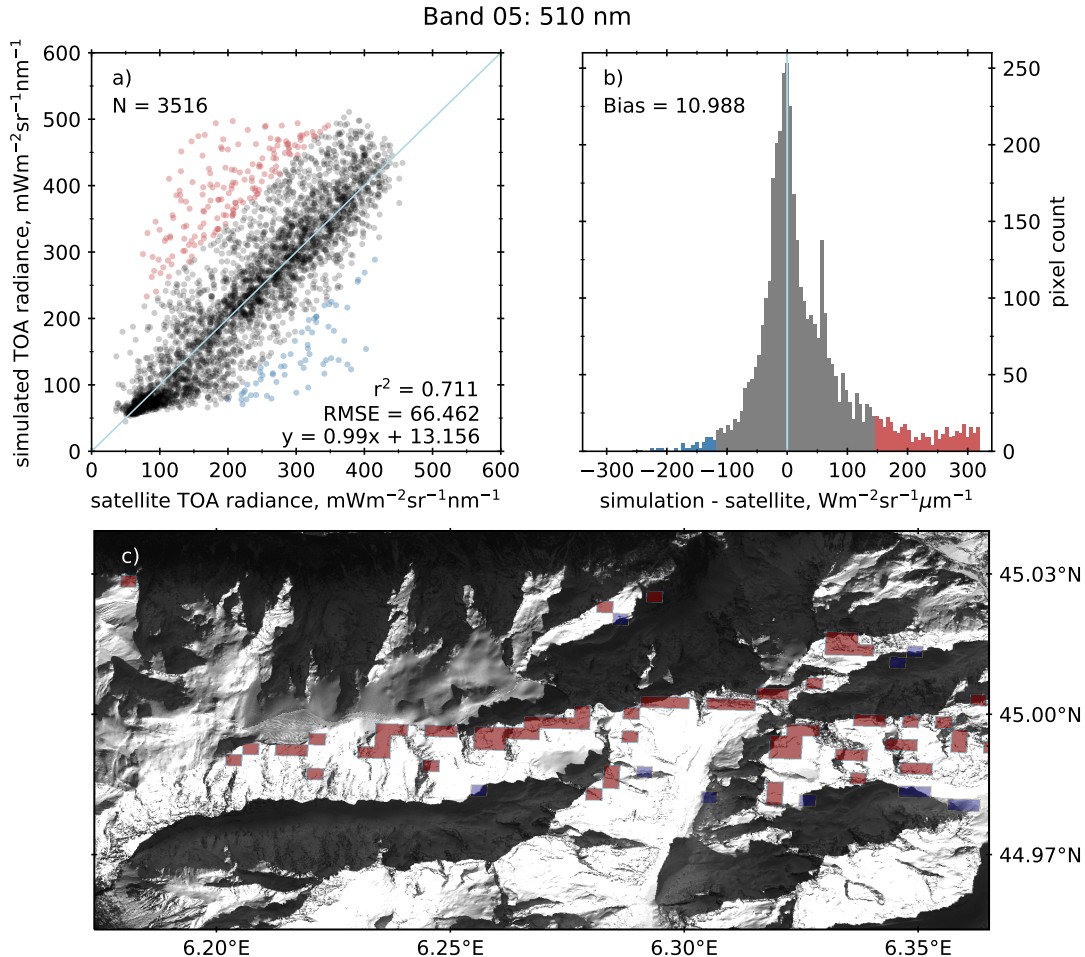

**Figure 5.** TOA radiance spectra computed with the rugged terrain model and observed with OLCI at 510 nm (Band 05) on 13 February 2018. a) Scatterplot of the relationship between the simulated TOA radiance and the concurrent satellite observations over the study area. The colored points (red and blue) represent under- or over-estimated by more than 2 $\sigma$ of the bias. b) difference between simulation and observation across the scene. c) Spatial distribution of the under- and over-estimated pixels shown on a 1.25m resolution SPOT 6 image, acquired on 19/02/2018 at 10h04m UTC via the Kalideos Alpes project (©AIRBUS DS).


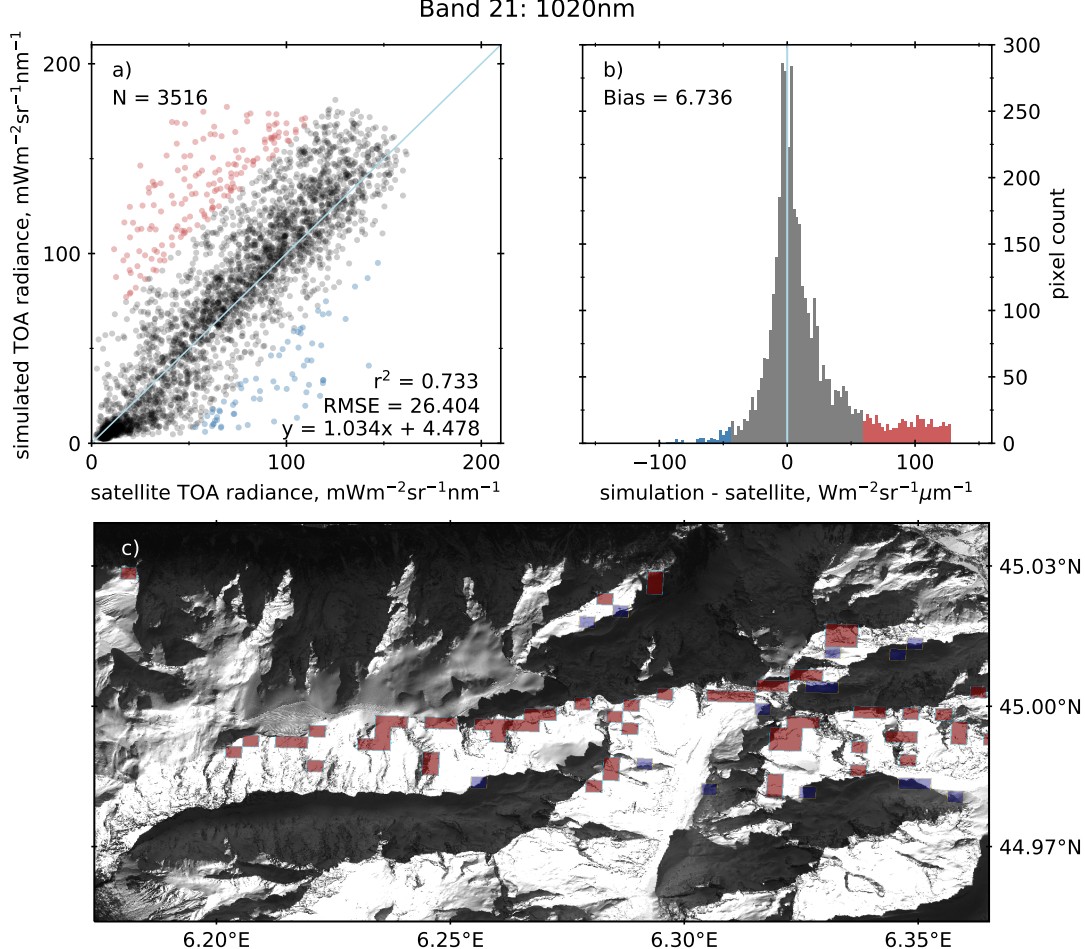

**Figure 6.** TOA radiance spectra computed with the rugged terrain model and observed with OLCI at 1020 nm (Band 21) on 13 February 2018. a) Scatterplot of the relationship between the simulated TOA radiance and the concurrent satellite observations over the study area. The colored points (red and blue) represent under- or over-estimated by more than 2 $\sigma$ of the bias. b) difference between simulation and observation across the scene. c) Spatial distribution of the under- and over-estimated pixels shown on a 1.25m resolution SPOT 6 image, acquired on 19/02/2018 at 10h04m UTC via the Kalideos Alpes project (©AIRBUS DS).



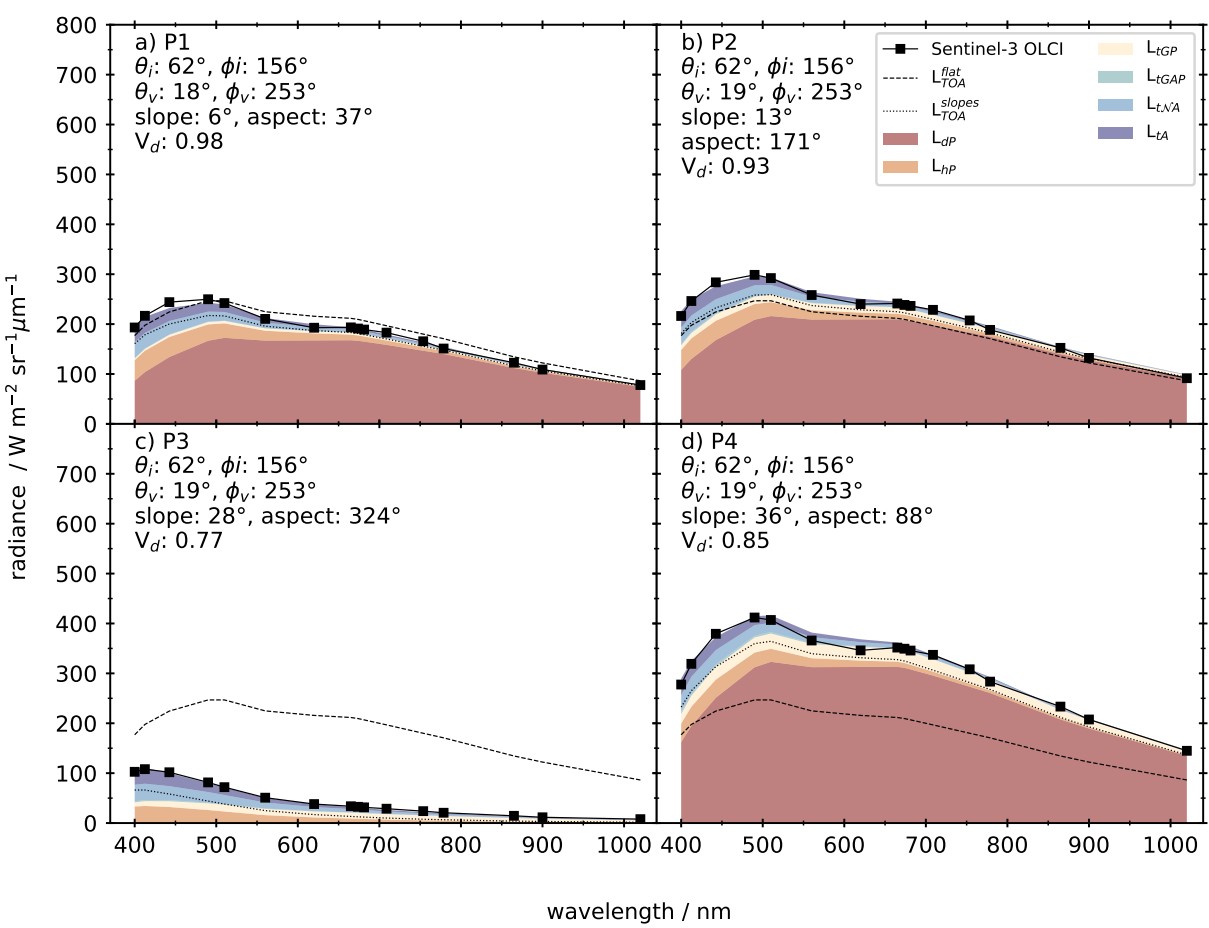

**Figure 7.** TOA radiance observed by Sentinel-3 OLCI (L1B) and simulated by REDRESS considering the full rugged terrain problem (stacked colored spectra), slopes only (dotted lines), and a flat surface (dashed lines), for four pixels with different terrain configurations on 13 February 2018. Note that values for $L_{tGAP}$ are too small to appear on the graph.

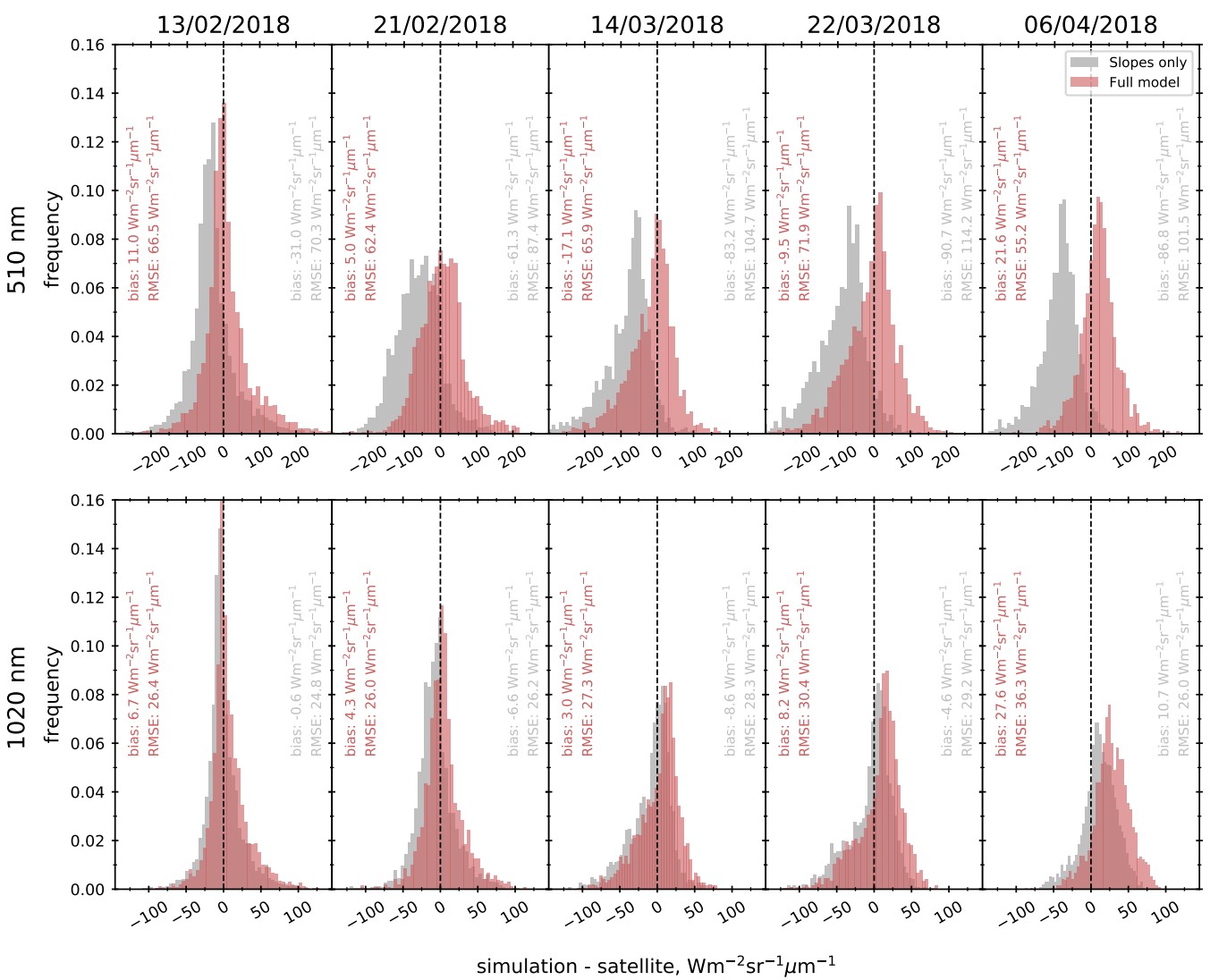

**Figure 8.** Distribution of the difference in TOA radiance between REDRESS simulations (red) and Sentinel-3 OLCI observations at Band 05 (510 nm, top) and Band 21 (1020 nm, bottom) for the five dates in 2018 over the study area. In gray, the same for simulations run considering slopes only. The bias and RMSE for REDRESS and REDRESS$_{slope}$ are indicated in each panel in red and grey respectively.



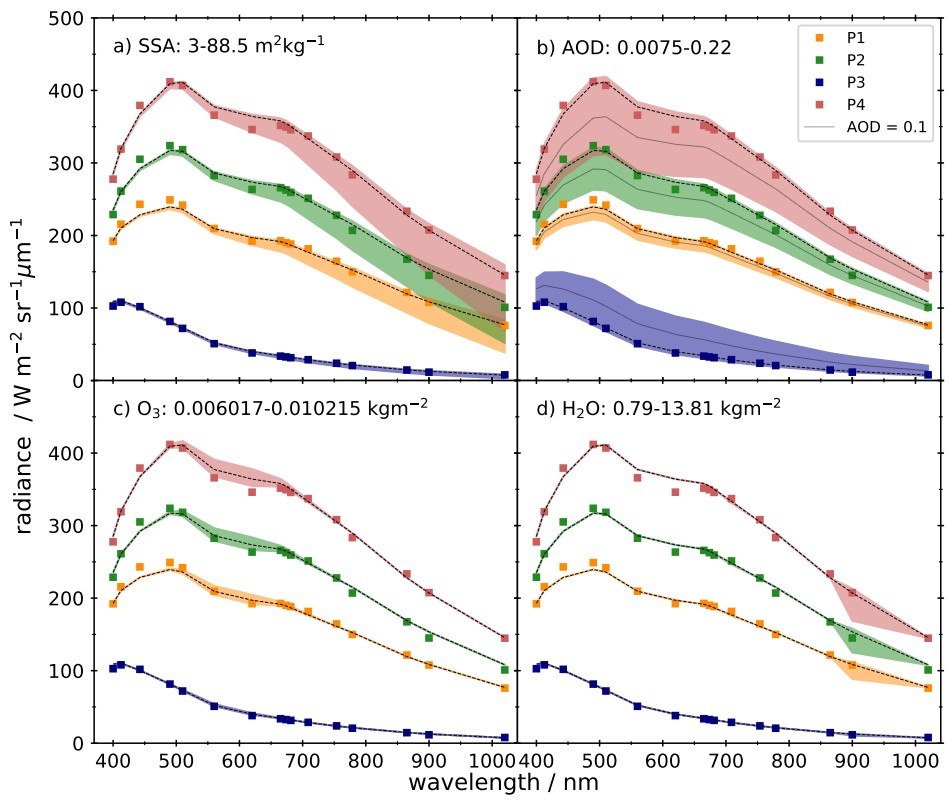

**Figure 9.** Sensitivity of the rugged terrain TOA radiance to the model input parameters for four pixels on 13 February 2018. Sentinel-3 OLCI L1B observations (colored squares) are compared to REDRESS simulations using the parameters described in Table 3 (dashed lines), and varying a) SSA input values, b) AOD values (for clarity an intermediate value of 0.1 is represented as a black line), c) atmospheric ozone column values, and d) atmospheric water vapour values (colored envelopes).



**Figure 10.** Modelled surface reflectance using the full rugged terrain (left) and slope-only (right) formulations on 13 February 2018 at 510 nm (top) and 1020 nm (bottom). Pixels containing less than 80% of snow were masked out (hatched region).


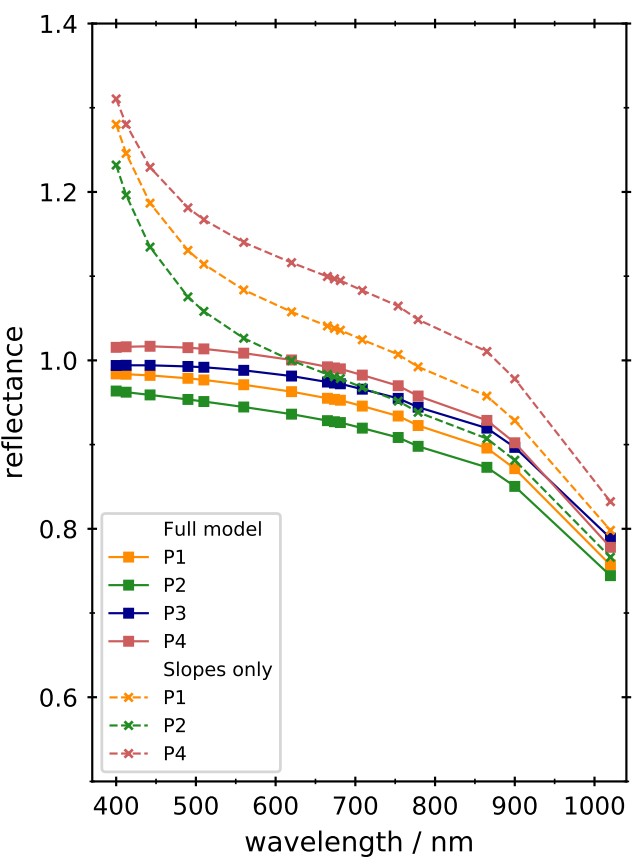

**Figure 11.** Reflectance spectra simulated for four pixels (P1, P2, P3, and P4) across the study site on 13 February 2018 using REDRESS (full lines) and REDRESS$_{slope}$ (dashed lines). The spectrum simulated for P3 with REDRESS$_{slope}$ is not shown here.



**Table 1.** Notation system for the different terms contributing to the TOA radiance measured by a satellite over rugged terrain. The notation for each term is made up of one or more letters from each category.

| Category | Notation | Description |
| --- | --- | --- |
| Flux direction | $E$ | Downwelling irradiance |
| | $L$ | Upwelling radiance |
| Source of the downwelling irradiance | $d$ | Direct radiation |
| | $h$ | Diffuse radiation |
| | $t$ | Total radiation |
| Observed surface | $P$ | Pixel under consideration |
| | $G$ | Ground surrounding $P$ |
| Contributions to the TOA signal | $\mathcal{N}$ | Neighbouring pixels (set radius) |
| | $A$ | Atmosphere |
| Surface configuration | flat | Horizontal surface, no topography |



**Table 2.** Summary of the four selected pixels across the study site.

| Point Name | Latitude [°] | Longitude [°] | Slope [°] | Aspect [°] | Location | Configuration |
|---|---|---|---|---|---|---|
| P1 | 45.06715 | 6.22100 | 6 | 37 | Emparis plateau | High elevation plateau |
| P2 | 45.04129 | 6.40925 | 13 | 171 | Col du Lautaret field site | Mountain pass surrounded by high peaks |
| P3 | 45.01883 | 6.41605 | 28 | 324 | South of Col du Lautaret | Steep North-facing slopes |
| P4 | 45.05602 | 6.42562 | 36 | 88 | North of Col du Lautaret | Steep South-facing slopes |





**Table 3.** Summary of the Sentinel-3A OLCI satellite acquisitions and the field measurements obtained for clear sky conditions during the 2017 / 2018 winter season. For each date the start and end times of the acquisition are indicated. The viewing and solar angles shown here were averaged across the scene. The average measured SSA across the transect is indicated with $\pm 1\sigma$ representing the spatial variability of the SSA across the site.

| Satellite acquisition date / time, UTC | Satellite viewing angles [°] | | Solar angles [°] | | N° of pixels | Atmospheric parameters | | | Number of field measurements | Measured SSA [m² kg⁻¹] | | |
|---|---|---|---|---|---|---|---|---|---|---|---|---|
| | $\theta_v$ | $\phi_v$ | $\theta_i$ | $\phi_i$ | | Water vapour column [$\mathrm{kg\,m^{-2}}$] | Ozone column [$\mathrm{kg\,m^{-2}}$] | Total AOD | | Average $\pm 1\sigma$ | Min | Max |
| 2018-02-13, 10:20:22 - 10h23:22 | 19.00 | 107.25 | 61.55 | 155.90 | 3516 | 1.75 | 0.008462 | 0.02 | 26 | 41.41 ± 1.75 | 25.38 | 88.55 |
| 2018-02-21, 10:12:53 - 10:15:53 | 8.31 | 105.99 | 59.41 | 153.08 | 2596 | 3.31 | 0.008276 | 0.15 | 12 | 33.66 ± 3.14 | 9.96 | 55.22 |
| 2018-03-14, 09:28:00 - 09:31:00 | 48.55 | 98.20 | 56.04 | 138.46 | 1894 | 2.30 | 0.008488 | 0.02 | 34 | 45.42 ± 9.70 | 28.34 | 69.16 |
| 2018-03-22, 09:20:31 - 09:23:31 | 54.21 | 96.91 | 53.89 | 135.22 | 1947 | 3.79 | 0.008830 | 0.05 | 38 | 27.37 ± 3.93 | 11.35 | 33.92 |
| 2018-04-06, 09:31:44 - 09:34:44 | 45.32 | 98.84 | 46.89 | 135.89 | 1894 | 5.12 | 0.007427 | 0.09 | 15 | 5.91 ± 1.33 | 3.06 | 8.15 |



**Table 4.** Range of values used to evaluate the sensitivity of the TOA simulations to the model input parameters.

| Input parameter | Value range |
| --- | --- |
| Snow SSA | $21.9$–$88.5 \, \mathrm{m^2 \, kg^{-1}}$ |
| AOD at 550 nm | $0.0075$–$0.2169$ |
| Total water vapour column | $0.79$–$13.81 \, \mathrm{kg \, m^{-2}}$ |
| Total ozone column | $0.006017$–$0.010215 \, \mathrm{kg \, m^{-2}}$ |