# Peer review of "Simulating Optical Top-Of-Atmosphere Radiance Satellite Images over Snow-Covered Rugged Terrain"

_The Cryosphere, 2020_

## Referee Comment (RC1) · Jeff Dozier (Referee) · 26 May 2020

The manuscript presents a comprehensive description and analysis of the reflection of solar radiation and its transmission back through the atmosphere to a satellite. My detailed comments mainly address clarification, but there may be an error in the equation for the view factor. Once made available, the REDRESS model will be a valuable contribution to snow science.

The paper perhaps understates the ways in which the model could be used to retrieve the snow properties that affect its albedo. The traditional approach presented here postulates that REDRESS could be inverted to retrieve the snow BRDF at the surface (the bottom of the atmosphere), and from those retrievals snow properties could be derived. Methods to estimate snow properties from MODIS (Painter et al., 2009) and Landsat (https://www.usgs.gov/land-resources/nli/landsat/landsat-fractional-snow-covered-area) use this approach.

However, Nolin and Dozier (2000) point out the difficulties this approach poses. Each step in the modeling process from calibration to atmospheric correction to accounting for the terrain introduces uncertainty and possibly error. An alternative, which this paper would nicely support, is instead to focus on attributes of the shape of the spectrum, hence my comments toward the end of the review about examining the spectral angle between the model and the measurements as a way to cut through some of these uncertainties that especially benefit when a continuous spectrum is available (Dozier et al., 2009).

Detailed comments:

Line 40: In this context, what does "surface-atmosphere coupling" mean? Multiple scattering between surface and atmosphere?

Lines 60-75: In this description of the difficulties, you should include surface roughness, on which you have already worked. A rough surface introduces the question of whether calculating the BRDF is necessary, given the uncertainty and subpixel heterogeneity in illumination and viewing angles. Also, for multispectral sensors, the signal convolves the spectral albedo (or BRDF) with the spectrum of the irradiance, which varies with atmospheric properties and the elevation of the surface.

Line 105: My reading of the AART model for snow is that its advantages lie in avoiding Mie scattering calculations. However, the computational burden of Mie scattering can be avoided by lookup tables. Some of the light-absorbing particles, both dust and soot, have traveled long distances and have sub-µm diameters. Calculating their scattering and absorption properties requires very careful programming: Code from Wiscombe (1980) works, whereas code based on Bohren and Huffman (2007) fails under some likely circumstances.

Line 112: "evaluate" not "evaluated"

Line 132: From the description, it appears that the viewing and illumination angles are for a flat surface. Please clarify here.

Lines 149-150: In reference to equations (3) and (4), $T_{dir} \uparrow$ and $T_{dir} \downarrow$ also depend on atmospheric properties.

Line 160, equation (8): Something seems wrong with this equation, and I do not see it in Sirguey (2009). The sentence following the equation defines $H(\phi)$ as a "horizon elevation," so if the pixel is flat and completely unobstructed, then does $H(\phi) = 0$? But then $V_d = 0$ from equation (8) where it should = 1. Figure 2 in Sirguey (2009) appears to define $H$ downward from zenith, as Dozier and Frew do. Moreover, Dozier and Frew (1990, eq 7b) incorporate slope and aspect into their

formulation for $V_d$, whereas Sirguey (2009) optionally incorporates slope and aspect into the complement of $V_d$, a "terrain configuration" factor (his equations 1 and 2). You must clarify and reconcile the definition of $H(\phi)$ and the corresponding correct equation for $V_d$.

Line 165, equation (9): How many iterations are needed? Does the equation converge?

Lines 170-180, equations (11) to (14): What are the assumptions about the shape of the surrounding terrain and its albedo? Dozier and Frew (1990) assume a bowl extending to the horizon in all directions. A viewshed would be more appropriate, but calculating a viewshed for every pixel doesn't take advantage of the fast horizon calculations (Dozier et al., 1981) that enable getting the horizon for every pixel in every direction. Therefore, we make some assumptions, please explain what they are in REDRESS, beyond what Line 184 says. Perhaps consider defining the assumptions *before* you present the equations.

Line 271, section 3.1.3: You should introduce and cite 6S at the beginning of this paragraph, rather than at the end.

Line 314, section 3.1.4: The quality of DEMs varies worldwide. Data from the Shuttle Radar Topography Mission (SRTM, Farr et al., 2007) are available nearly everywhere between 60°N/S. GDEM data (https://earthdata.nasa.gov/learn/articles/new-aster-gdem) from ASTER have slightly lesser coverage but extend to higher latitudes. Finer resolution DEMs from photogrammetry from aircraft or fine-resolution satellite imagery (Shean et al., 2016) are available in some regions. At the finest scale, DEMs from airborne lidar instruments are used in some mountainous areas (Painter et al., 2016; Trujillo et al., 2007),  and terrestrial scanning lidar (Deems et al., 2013) and structure-from-motion analyses of imagery from small drones have provided topographic information at very fine scales (Fonstad et al., 2013). The point though is that the differencing operations that are needed to calculate the illumination and viewing geometry introduce noise or, if the calculations are filtered, smooth the calculations. Moreover, at the scales of Sentinel-3, Sentinel-2, Landsat 8, MODIS, and VIIRS, topographic variability occurs within the pixel. Because of these limitations, one must be cautious about the accuracy, precision, and internal heterogeneity of calculations of angles $\tilde{\theta}_{i,v}$ and $\tilde{\phi}_{i,v}$. This section should address those limitations, particularly in how they affect the BRDF estimates.

Line 352: "given that the model considers a fixed SSA value across the scene" seems like an unnecessary constraint.

Around Line 360: How are the atmospheric parameters from CAMS adjusted for surface elevation? I am not familiar with the product, but I hazard each 0.4° cell has an elevation associated with it. From that, you could use some sort of pressure weighting scheme to estimate water vapor, ozone, and aerosol optical depth pixel-by-pixel (Bair et al., 2016; Rittger et al., 2016).

Line 366: The model apparently considers clean snow only. Make this clear upfront. Given that constraint, would the difference between clean and dirty snow wash out some of the details about, for example, multiple reflections?

Line 380, equation (27): Display the equation in a way that makes clear the position of the second term on the right.

Line 399: Statements such as "an excellent agreement between the measured and modelled TOA radiance is observed at both wavelengths" are unsatisfactory. Use the metrics presented in the following paragraph (RSME, bias, etc) to characterize the relationship, rather than an adjective.

Lines 416-421: Refer to my earlier comments (section 3.1.4) about DEMs generally. For the 1 arc sec (~30 m) DEMs from SRTM or GDEM, you're kidding yourself if you invest too heavily in the accuracy of the illumination geometry.  While elevation itself is mostly a continuous variable, illumination angle is not. The discussion should separate uncertainties in REDRESS vs. those in the input data.

Lines 440-450: the section heading (4.1.2 Spectral performance) misleads a bit, as just 2 wavelengths are presented. The idea of a spectrometer is that the shape of the spectrum enables analysis; a spectrometer is not just a multispectral sensor with lots of bands. Therefore, a useful addition would include information about how well the model matches the spectrometer. How does the Euclidean norm of the residuals between measurement and model vary across the landscape? What about the spectral angle?

$$Norm = \left\| \overrightarrow{L_{mod}} - \overrightarrow{L_{meas}} \right\|_2$$

$$\cos \sphericalangle = \frac{\overrightarrow{L_{mod}} \cdot \overrightarrow{L_{meas}}}{\left\| \overrightarrow{L_{mod}} \right\|_2 \times \left\| \overrightarrow{L_{meas}} \right\|_2}$$

Lines 605-620: The discussion about the quality of the DEM is insightful. One issue not mentioned though is that although the estimation of the illumination and viewing geometry improve as the pixel coarsens in comparison to the resolution of the DEM, the subpixel heterogeneity in the topography becomes more problematic. Perhaps mention that?

Lines 625-630: Indeed the quality of the knowledge about the atmosphere is important, but so is sensor calibration. The paper is already long, so I avoid asking you to address the effect of uncertainty and drift in calibration, but at least mention it.

Table 2. Indicate that Aspect is measured clockwise from North. This is the common convention, although it is inconsistent with a right-hand coordinate system. When I started working on topographic radiation problems, my go-to text was *Physical Climatology* (Sellers, 1965), with Aspect 0° to the South, positive east, negative west (as we use for longitude). Clockwise-from-North is most common, but not universal, hence the need to specify.

Citations in the review

Bair, E. H., Rittger, K., Davis, R. E., Painter, T. H., and Dozier, J.: Validating reconstruction of snow water equivalent in California's Sierra Nevada using measurements from the NASA Airborne Snow Observatory, Water Resources Research, 52, 8437-8460, https://doi.org/10.1002/2016WR018704, 2016.

Bohren, C. F., and Huffman, D. R.: Absorption and Scattering of Light by Small Particles, John Wiley and Sons, New York, 530 pp., 2007.

Deems, J. S., Painter, T. H., and Finnegan, D. C.: Lidar measurement of snow depth: A review, Journal of Glaciology, 59, 467-479, https://doi.org/10.3189/2013JoG12J154, 2013.

Dozier, J., and Frew, J.: Rapid calculation of terrain parameters for radiation modeling from digital elevation data, IEEE Transactions on Geoscience and Remote Sensing, 28, 963-969, https://doi.org/10.1109/36.58986, 1990.

Dozier, J., Bruno, J., and Downey, P.: A faster solution to the horizon problem, Computers and Geosciences, 7, 145-151, https://doi.org/10.1016/0098-3004(81)90026-1, 1981.

Dozier, J., Green, R. O., Nolin, A. W., and Painter, T. H.: Interpretation of snow properties from imaging spectrometry, Remote Sensing of Environment, 113, S25-S37, https://doi.org/10.1016/j.rse.2007.07.029, 2009.

Farr, T. G., Rosen, P. A., Caro, E., Crippen, R., Duren, R., Hensley, S., Kobrick, M., Paller, M., Rodriguez, E., Roth, L., Seal, D., Shaffer, S., Shimada, J., Umland, J., Werner, M., Oskin, M., Burbank, D., and Alsdorf, D.: The Shuttle Radar Topography Mission, Reviews of Geophysics, 45, RG2004, https://doi.org/10.1029/2005RG000183, 2007.

Fonstad, M. A., Dietrich, J. T., Courville, B. C., Jensen, J. L., and Carbonneau, P. E.: Topographic structure from motion: a new development in photogrammetric measurement, Earth Surface Processes and Landforms, 38, 421-430, https://doi.org/10.1002/esp.3366, 2013.

Nolin, A. W., and Dozier, J.: A hyperspectral method for remotely sensing the grain size of snow, Remote Sensing of Environment, 74, 207-216, https://doi.org/10.1016/S0034-4257(00)00111-5, 2000.

Painter, T. H., Rittger, K., McKenzie, C., Slaughter, P., Davis, R. E., and Dozier, J.: Retrieval of subpixel snow-covered area, grain size, and albedo from MODIS, Remote Sensing of Environment, 113, 868-879, https://doi.org/10.1016/j.rse.2009.01.001, 2009.

Painter, T. H., Berisford, D. F., Boardman, J. W., Bormann, K. J., Deems, J. S., Gehrke, F., Hedrick, A., Joyce, M., Laidlaw, R., Marks, D., Mattmann, C., McGurk, B., Ramirez, P., Richardson, M., Skiles, S. M., Seidel, F. C., and Winstral, A.: The Airborne Snow Observatory: Fusion of scanning lidar, imaging spectrometer, and physically-based modeling for mapping snow water equivalent and snow albedo, Remote Sensing of Environment, 184, 139-152, https://doi.org/10.1016/j.rse.2016.06.018, 2016.

Rittger, K., Bair, E. H., Kahl, A., and Dozier, J.: Spatial estimates of snow water equivalent from reconstruction, Advances in Water Resources, 94, 345-363, https://doi.org/10.1016/j.advwatres.2016.05.015, 2016.

Sellers, W. D.: Physical Climatology, University of Chicago Press, Chicago, 272 pp., 1965.

Shean, D. E., Alexandrov, O., Moratto, Z. M., Smith, B. E., Joughin, I. R., Porter, C., and Morin, P.: An automated, open-source pipeline for mass production of digital elevation models (DEMs) from very-high-resolution commercial stereo satellite imagery, ISPRS Journal of Photogrammetry and Remote Sensing, 116, 101-117, https://doi.org/10.1016/j.isprsjprs.2016.03.012, 2016.

Sirguey, P.: Simple correction of multiple reflection effects in rugged terrain, International Journal of Remote Sensing, 30, 1075-1081, https://doi.org/10.1080/01431160802348101, 2009.

Trujillo, E., Ramírez, J. A., and Elder, K. J.: Topographic, meteorologic, and canopy controls on the scaling characteristics of the spatial distribution of snow depth fields, Water Resources Research, 43, W07409, https://doi.org/10.1029/2006WR005317, 2007.

Wiscombe, W. J.: Improved Mie scattering algorithms, Applied Optics, 19, 1505-1509, https://doi.org/10.1364/AO.19.001505, 1980.

---

## Referee Comment (RC2) · Anonymous Referee #2 · 15 Jul 2020

In my opinion the manuscript is outstanding: it contains a sound description of the theoretical background, a description of the parameters (snow, atmosphere, topography) and a sensitivity analysis.

The field measurements (SSA along transects) during the Sentinel-3 overpass are particularly valuable, since they are used to compare simulated results with scene observations.

I have only a few suggestions to improve the manuscript:

- line 148: $E_0$ extraterrestrial solar irradiance: which model is used (Thuillier 2003) ?

[Figure]

- the up/down arrow Tdir(arrow_up or arrow_down) should be a superscript to Tdir (arrow not standing alone) (concerns all equations and in the running text)

- line 328: inferior to 2 days better: less than 2 days

- Fig. 4: right side: the Radiance unit W m-1 sr -1 $\mu$m-1 –> W m-2 sr-1 $\mu$m-1

Question concerning the visual appearance: the study area is 14 km x 18 km, i.e.46 x 60 pixels, so I would expect more blocky structures. Was a certain histogram-stretch applied to make it look smoother?

- Fig. 5 and 6: use of the radiance unit [mW m-2 sr-1 nm-1] (plot ordinate and left scattterplot), which is correct, but I suggest to use [W m-2 sr-1 $\mu$m-1] as done in the text and in the right plot of the histogram.

---

## Author Comment (AC1) · 31 Aug 2020

**Answer to Anonymous Referee 2**

The authors would like to thank Anonymous Referee 2 for their general and supportive analysis of the paper, as well as for the detailed comments, which we have taken into account. The reviewer's initial comments are reported in blue, and our answers are written in black.

- line 148: E0 extraterrestrial solar irradiance: which model is used (Thuillier 2003) ?

The EO extraterrestrial solar irradiance is obtained from the 6S model, which uses Neckel and Labs, 1984. We have added the reference line 155 (of the revised version of the manuscript).

- the up/down arrow Tdir(arrow_up or arrow_down) should be a superscript to Tdir (arrow not standing alone) (concerns all equations and in the running text)

The annotations $T_{dir} \downarrow$ and $T_{dir} \uparrow$ have been changed to $T_{dir}^{\downarrow}$ and $T_{dir}^{\uparrow}$ respectively throughout the manuscript.

- line 328: inferior to 2 days better: less than 2 days

Corrected.

- Fig. 4: right side: the Radiance unit W m-1 sr -1 $\mu$m-1 → W m-2 sr-1 $\mu$m-1

Corrected.

Question concerning the visual appearance: the study area is 14 km x 18 km, i.e.46 x 60 pixels, so I would expect more blocky structures. Was a certain histogram-stretch applied to make it look smoother?

We would like to thank the reviewer for this comment as it allowed us to correct a typo in the study area size. The study area size is $\approx 17 \times 20$ km and not $\approx 14 \times 18$ km as stated. This has been corrected in the manuscript. Nevertheless, the image size is similar ($58 \times 67$ pixels rather than $46 \times 60$ pixels). The apparent smoothness of the image in the figure may come from the rendering of the (small) figure in QGIS, as when the image is viewed at 100% it appears more

blocky.

- Fig. 5 and 6: use of the radiance unit [mW m-2 sr-1 nm-1] (plot ordinate and left scattterplot), which is correct, but I suggest to use [W m-2 sr-1 $\mu$m-1] as done in the text and in the right plot of the histogram.

We apologise for this lack of consistency and have modified the unit, that now reads: W m$^{-2}$ sr$^{-1}\mu$ m $^{-1}$.

REFERENCES:

H. Neckel, and D. Labs, The solar radiation between 3300 and 12500, Solar Physics, 90, 205- 258, 1984.

---

## Author Comment (AC2) · 31 Aug 2020

**Answer to Referee 1**

The reviewer's initial comments are reported in blue, our answers are written in black, and the corrections in the paper are highlighted in red. The line numbers referred to in this document correspond to the line numbers in the revised manuscript.

**General comments**

The manuscript presents a comprehensive description and analysis of the reflection of solar radiation and its transmission back through the atmosphere to a satellite. My detailed comments mainly address clarification, but there may be an error in the equation for the view factor. Once made available, the REDRESS model will be a valuable contribution to snow science. The paper perhaps understates the ways in which the model could be used to retrieve the snow properties that affect its albedo. The traditional approach presented here postulates that REDRESS could be inverted to retrieve the snow BRDF at the surface (the bottom of the atmosphere), and from those retrievals snow properties could be derived. Methods to estimate snow properties from MODIS (Painter et al., 2009) and Landsat (`https://www.usgs.gov/land-resources/nli/landsat/landsat-fractional-snow-covered-area`) use this approach. However, Nolin and Dozier (2000) point out the difficulties this approach poses. Each step in the modeling process from calibration to atmospheric correction to accounting for the terrain introduces uncertainty and possibly error. An alternative, which this paper would nicely support, is instead to focus on attributes of the shape of the spectrum, hence my comments toward the end of the review about examining the spectral angle between the model and the measurements as a way to cut through some of these uncertainties that especially benefit when a continuous spectrum is available (Dozier et al., 2009).

We would like to thank Jeff Dozier for the great attention shown in reading and commenting our work, as well as the interesting discussion points that have helped us improve the paper. Our approach in this paper was to provide a pedagogical presentation of the results, in such a way that effects of rugged terrain on reflectance retrievals would be easy to understand despite the complex equations used in the model (that are of interest for more advanced users). This simplification of the results entails limiting us to showing a limited number of points in the study area, or a few wavelengths in the case of our maps. The approach makes the results easier to comprehend for remote sensing specialists

interested in snow in mountainous areas compared to the alternative proposed by the reviewer. However, the general comments above are of great interest, and we have addressed them further down in the detailed comments. We agree that focusing on the shape of the spectrum is an interesting alternative approach and will consider this point in further work.

**Detailed comments:**

Line 40: In this context, what does "surface-atmosphere coupling" mean? Multiple scattering between surface and atmosphere?

Indeed, in this case the "surface-atmosphere coupling" term is used to describe multiple scattering events between the surface and the atmosphere. For clarity, the sentence was rephrased as follows (Line 38):

For modelling purposes, the radiative fluxes contributing to the top-of-atmosphere (TOA) radiance over a mountainous scene can be broken down into different terms (Lenot et al., 2009), where the downwelling fluxes are split into four terms: direct, diffuse, reflections from neighbouring slopes, and multiple scattering between the surface and the atmosphere, hereinafter referred to as surface-atmosphere coupling.

Lines 60-75: In this description of the difficulties, you should include surface roughness, on which you have already worked. A rough surface introduces the question of whether calculating the BRDF is necessary, given the uncertainty and subpixel heterogeneity in illumination and viewing angles. Also, for multi-spectral sensors, the signal convolves the spectral albedo (or BRDF) with the spectrum of the irradiance, which varies with atmospheric properties and the elevation of the surface.

We agree that small-scale surface roughness is a factor that can affect retrievals of snow physical properties, as we have pointed out in previous studies (Larue et al., 2020). The following sentences have been added to the manuscript (Line 78), and this point has also been addressed in the discussion (see comment further down):

All of the aforementioned models assume that the surface of the snowpack in each pixel is smooth and thus neglect macroscopic surface roughness. Yet observations have shown that for low sun angles, which can occur even at solar noon due to steep slopes, surface roughness causes a decrease in albedo compared to a smooth surface with the same properties (Larue et al., 2020). Furthermore, pronounced roughness features that are not resolved by the DEM (e.g. figure 2 in Guyomarc'h et al., 2019), introduce uncertainty in the calculations of surface reflectance by causing sub-pixel variability in the pixel's illumination and viewing angles. Therefore these physical models are expected to perform less well over rough snow surfaces.

Line 105: My reading of the AART model for snow is that its advantages

lie in avoiding Mie scattering calculations. However, the computational burden of Mie scattering can be avoided by lookup tables. Some of the light-absorbing particles, both dust and soot, have traveled long distances and have sub-$\mu$m diameters. Calculating their scattering and absorption properties requires very careful programming: Code from Wiscombe (1980) works, whereas code based on Bohren and Huffman (2007) fails under some likely circumstances.

One of the reasons we used the AART model for snow, other than for its fast computational abilities, is that the formulation accounts for non-spherical snow particles. Although the shape of snow "grains" cannot be explicitly specified in the model, they are controlled by two free parameters: the absorption enhancement parameter $B$ and the asymmetry parameter, $g$ (Libois et al., 2014). In this study, we used values of $B$=1.6 and $g$=0.85 as recommended by Libois et al., 2014 and Tuzet et al., 2019.

Although we only consider clean snow in this study, the AART theory has been used to model snow containing black carbon or mineral aerosol deposits (Kokhanovsky et al., 2018), and this addition could be envisaged in future work.

Line 112: "evaluate" not "evaluated"

Corrected.

Line 132: From the description, it appears that the viewing and illumination angles are for a flat surface. Please clarify here.

This is correct. We modified the manuscript accordingly :
$\theta_i$, $\theta_v$, $\phi_i$, and $\phi_v$ describe the illumination and viewing zenith and azimuth angles respectively for a flat surface.

Lines 149-150: In reference to equations (3) and (4), $\mathrm{T}_{dir} \uparrow$ and $\mathrm{T}_{dir} \downarrow$ depend on atmospheric properties.

We added the clarification to the text. Line 156 now reads:
As is the case for $T_{dir}^{\uparrow}(P, \theta_v, \phi_v)$, $T_{dir}^{\downarrow}(P, \theta_i, \phi_i)$ depends on the location and altitude of the pixel P, as well as on the atmospheric properties.

Line 160, equation (8): Something seems wrong with this equation, and I do not see it in Sirguey (2009). The sentence following the equation defines H($\phi$) as a "horizon elevation," so if the pixel is flat and completely unobstructed, then does H($\phi$) = 0? But then V$_d$ = 0 from equation (8) where it should = 1. Figure 2 in Sirguey (2009) appears to define H downward from zenith, as Dozier and Frew do. Moreover, Dozier and Frew (1990, eq 7b) incorporate slope and aspect into their formulation for V$_d$ whereas Sirguey (2009) optionally incorporates slope and aspect into the complement of V$_d$, a "terrain configuration" factor (his equations 1 and 2). You must clarify and reconcile the definition of H($\phi$) and the corresponding correct equation for V$_d$.

We agree that there is a discrepancy between the definition of H($\phi$) and the corresponding equation for $V_d$ in Section 2.1.2 of the manuscript and would like to thank the reviewer for pointing it out. After investigation, it appears that the equation written in the paper does not correspond to the one in the code, the incorrect formulation being in the paper. In the model, we used the formulation from equation 7b in Dozier and Frew, 1990, and the horizon values (calculated as "elevation from horizontal") are converted to "downward from zenith" in the routine. It should be noted that Sirguey et al., 2009 wrongly calculated the sky-view factor, as Pascal Sirguey stated later in his thesis (p.109 of Sirguey, 2009), which may lead to some confusion. In ModImLab's code, the version of $V_d$ from Dozier and Frew, 1990 is also implemented. We have corrected equation 8 so that it reflects the calculations performed in the code. Line 165 is now written as follows:

with $E_{hP}^{\text{flat}}(P, \theta_i, \phi_i)$ the irradiance received by a theoretically horizontal surface modulated by $V_d(P)$, the sky-view factor (Dozier et al., 1981) varying from 0 to 1, and approximated by Dozier and Frew, 1990 as:

$$V_d = \frac{1}{2\pi} \int_0^{2\pi} \Big( \cos\theta_n \sin^2 H_z(\phi) + \sin\theta_n \cos(\phi - \phi_n) \\ (H_z(\phi) - \sin H_z(\phi) \cos H_z(\phi)) \Big) d\phi, \tag{1}$$

for which the horizon angle from zenith $H_z(\phi)$ for a given azimuth $\phi$ is converted as $H_z(\phi) = H(\phi) - 90$ from the horizon elevation $H(\phi)$, itself calculated using Dozier et al. (1981)'s algorithm.

Line 165, equation (9): How many iterations are needed? Does the equation converge?

We have added the following sentence after equation 9 (Line 174):
Numerical tests over different terrain configurations in the French Alps (not presented here) have shown that the equation converges in approximately 4–6 iterations.

Lines 170-180, equations (11) to (14): What are the assumptions about the shape of the surrounding terrain and its albedo? Dozier and Frew (1990) assume a bowl extending to the horizon in all directions. A viewshed would be more appropriate, but calculating a viewshed for every pixel doesn't take advantage of the fast horizon calculations (Dozier et al., 1981) that enable getting the horizon for every pixel in every direction. Therefore, we make some assumptions, please explain what they are in REDRESS, beyond what Line 184 says. Perhaps consider defining the assumptions before you present the equations.

We have taken into account the suggestion to define the assumptions before presenting the equations, and have added information about the assumptions made about the shape of surrounding terrain. We have modified Section 2.1.2 so that at line 175 appears:

When calculating the contributions from the surrounding terrain to an observed pixel several assumptions are made in the following equations. First, it is assumed that all the pixels in the neighbourhood $\mathcal{N}$ of a pixel P are receiving the same irradiance as a horizontal surface. Second, the shape of the terrain surrounding the pixel P is considered as in Dozier and Frew (1990), in that the terrain forms a bowl extending to the horizon in all azimuths ($\phi$). Lastly, the surface reflectance of pixels surrounding the observed pixel is assumed to lambertian for simplicity, thus $R(M, \theta_i^M, \theta_v^M, \phi_i^M, \phi_v^M) = S(M)$ (see equation 12).

Subsequently, the text following equation 15 was removed to avoid repetitions.

Line 271, section 3.1.3: You should introduce and cite 6S at the beginning of this paragraph, rather than at the end.

Section 3.1.3 was modified to introduce and cite 6S at the beginning of the paragraph. Line 286 now reads:

The atmospheric components are calculated using the 6S radiative transfer model (Vermote et al., 1997) that is initialised with four main parameters: water vapour content, the total ozone column, the type of aerosol present in the atmosphere and the total aerosol optical depth (AOD) obtained from the datasets described in Section 3.2.4.

Line 293 was also modified:

In the current setup, REDRESS, written in python, uses the Py6S module (Wilson, 2013) to run the 6S Fortran code, but the model is designed for an easy implementation of other radiative transfer codes.

Line 314, section 3.1.4: The quality of DEMs varies worldwide. Data from the Shuttle Radar Topography Mission (SRTM, Farr et al., 2007) are available nearly everywhere between 60°N/S. GDEM data (`https://earthdata.nasa.gov/learn/articles/new-aster-gdem`) from ASTER have slightly lesser coverage but extend to higher latitudes. Finer resolution DEMs from photogrammetry from aircraft or fine-resolution satellite imagery (Shean et al., 2016) are available in some regions. At the finest scale, DEMs from airborne lidar instruments are used in some mountainous areas (Painter et al., 2016; Trujillo et al., 2007), and terrestrial scanning lidar (Deems et al., 2013) and structure-from-motion analyses of imagery from small drones have provided topographic information at very fine scales (Fonstad et al., 2013). The point though is that the differencing operations that are needed to calculate the illumination and viewing geometry introduce noise or, if the calculations are filtered, smooth the calculations. Moreover, at the scales of Sentinel-3, Sentinel-2, Landsat 8, MODIS, and VIIRS, topographic variability occurs within the pixel. Because of these limitations, one must be cautious about the accuracy, precision, and internal heterogeneity of calculations of angles $\tilde{\theta}_{i,v}$ and $\tilde{\phi}_{i,v}$. This section should address those limitations, particularly in how they affect the BRDF estimates.

We agree with the point raised about the effects of sub-pixel topographic variability on the BRDF estimates. The reviewer's comment focuses on the limitations of the DEM-based products and we suggest addressing this valid point in the discussion section. We have therefore modified Section 5.2 (Limitations and further improvements) to account for the comment above. For other changes in the discussion concerning the products calculated from the DEM, please refer to the reviewers comment further down (page 9 of this review). Furthermore, the changes we have applied reflect in part the reviewers previous comment on the BRDF. In the revised manuscript, we have added the following text (Line 633):

However, using DEM products with a higher resolution than the satellite image, as is recommended by Richter(1998) who suggest using a DEM with a resolution of 0.25 times the satellite image size, also introduces uncertainties in the calculation of topographic parameters at the pixel scale, that in turn affect the retrieval of snow properties. The use of high-resolution DEMs that are more and more accessible thanks to widespread high-resolution satellite imagery, airborne or drone-based platforms (e.g. Nolan et al., 2015; Bühler et al., 2016; Deschamps-Berger et al., 2020), may not necessarily improve the model's results. Indeed, the higher-resolution products calculated from the DEM, including illumination and viewing geometries, need to be resampled to the coarser satellite pixel, causing a smoothing of angles. In addition, at the scale of hundreds of meters, the sub-pixel heterogeneity in terms of topography might be high. The BRDF of the snow surface that is calculated using these smooth geometries may be not be representative of the surface, as rougher surfaces tend to smooth out the strong anisotropy of snow (e.g Warren et al., 1998). Therefore, the BRDF model used in this study, that considers a smooth surface, may have a tendency to produce an excessively pronounced signal compared to the one measured by the satellite. Further work on the accurate representation of the terrain at different spatial scales is thus recommended.

Line 352: "given that the model considers a fixed SSA value across the scene" seems like an unnecessary constraint.

We acknowledge that the statement is unclear and have reformulated it. Line 365 was changed to:

The average SSA measured along the transect was used as a single input SSA value, given that in the current model setup, snow is described using a fixed SSA value across the scene.

Around Line 360: How are the atmospheric parameters from CAMS adjusted for surface elevation? I am not familiar with the product, but I hazard each 0.4° cell has an elevation associated with it. From that, you could use some sort of pressure weighting scheme to estimate water vapor, ozone, and aerosol optical depth pixel-by-pixel (Bair et al., 2016; Rittger et al., 2016).

To the author's knowledge, the CAMS atmospheric parameters are already

adjusted for surface elevation. The surface products were selected for this study (when downloading the data from the ECMWF catalog, the user can select the "surface", a "pressure level" or a "model level"). The CAMS NRT product is based on a combination of data from the ECMWF IFS model and data assimilation. The principle source of surface elevation data in IFS is SRTM30 between 60°N and 60°S (https://www.ecmwf.int/en/elibrary/16648-part-iv-physical-processes).

Line 366: The model apparently considers clean snow only. Make this clear upfront. Given that constraint, would the difference between clean and dirty snow wash out some of the details about, for example, multiple reflections?

To make this clearer to the reader early on, the following statement was added at the end of Section 3.2.1 (line 283 in the revised manuscript):

It should also be noted that in the current version of REDRESS, snow is considered to be free of impurities, such as black carbon or mineral aerosol deposits.

Line 380, equation (27): Display the equation in a way that makes clear the position of the second term on the right.

The layout of the equation was revised, and is presented as follows in the updated manuscript

$$R(P, \theta_v, \phi_v, \theta_i, \phi_i) = \frac{\Big(L_{\text{TOA}}(P, \theta_i, \theta_v, \phi_i, \phi_v) - L_{tA}(\theta_i, \theta_v, \phi)\Big)\pi}{T_{dir}^{\uparrow}(P, \theta_v, \phi_v)\Phi(\tilde{\theta}_v, \tilde{\phi}_v)\Big(E_{dP}(P, \tilde{\theta}_i, \tilde{\phi}_i) + E_{hP}(P, \tilde{\theta}_i, \tilde{\phi}_i)\Big)}$$

$$(2)$$

Line 399: Statements such as "an excellent agreement between the measured and modelled TOA radiance is observed at both wavelengths" are unsatisfactory. Use the metrics presented in the following paragraph (RSME, bias, etc) to characterize the relationship, rather than an adjective.

We apologise for this vague statement and have corrected the sentence (Line 413):

A correlation larger than 0.7 associated with low bias ($5$–$10\,\text{Wm}^{-2}\text{sr}^{-1}\mu\text{m}^{-1}$) between the measured and modelled TOA radiance is observed at both wavelengths, highlighting the model's ability to reproduce the large variations in TOA radiance across the scene despite the same snow intrinsic properties being applied to all pixels.

Lines 416-421: Refer to my earlier comments (section 3.1.4) about DEMs generally. For the 1 arc sec ( 30 m) DEMs from SRTM or GDEM, you're kidding yourself if you invest too heavily in the accuracy of the illumination geometry. While elevation itself is mostly a continuous variable, illumination angle is not.

The discussion should separate uncertainties in REDRESS vs. those in the input data.

Although we acknowledge that it would be misleading to focus entirely on the accuracy of the illumination geometry and shadows with a 1 arc sec DEM, we found it relevant and of interest to show how the largest discrepancies between the model and the satellite observations are spatially distributed. The results show how important the characterisation of the terrain and the scale-factor between the DEM and the satellite image have an impact of the calculations, even though a higher-resolution DEM doesn't solve all the problems as pointed out by the reviewer. We feel that the sentence line 414 in the original manuscript was slightly misleading and have updated line 430 (in the revised manuscript) to read:

To identify the spatial distribution of the pixels for which the bias between the model and the satellite observations was the highest, the values over- and under-estimated by more than 2 standard deviations of the bias were colored in red and blue respectively, and identified as such in all the panels of Figures 5 and 6.

Lines 440-450: the section heading (4.1.2 Spectral performance) misleads a bit, as just 2 wavelengths are presented. The idea of a spectrometer is that the shape of the spectrum enables analysis; a spectrometer is not just a multi-spectral sensor with lots of bands. Therefore, a useful addition would include information about how well the model matches the spectrometer. How does the Euclidean norm of the residuals between measurement and model vary across the landscape? What about the spectral angle?

$$Norm = ||\overrightarrow{L_{mod}} - \overrightarrow{L_{meas}}||_2$$
$$\cos \measuredangle = \frac{\overrightarrow{L_{mod}} \cdot \overrightarrow{L_{meas}}}{||\overrightarrow{L_{mod}}||_2 \times ||\overrightarrow{L_{meas}}||_2}$$

By "Spectral performance" we meant to present the comparison between the model output and the satellite observations for all relevant OLCI bands, as opposed to only two wavelengths as presented in the previous section (Spatial performance). To be less mis-leading we have renamed Sections 4.1.1 and 4.1.2 "4.1.1 Spatial comparison at two wavelengths" and "4.1.2 Band-wise comparison at four locations" respectively. We also investigated the suggested metrics of Euclidean norm and spectral angle for the 13th February 2018 (see Figure 1). We find that the figure conveys a similar message as the current manuscript and to keep the paper as accessible as possible and because of its length and high number of figures, we suggest not to add it to the revised manuscript.

[Figure]

Figure 1: Left: Euclidean norm of the residuals between measurement and model for the 13th of February 2018. Right: Id. for the spectral angle.

Lines 605-620: The discussion about the quality of the DEM is insightful. One issue not mentioned though is that although the estimation of the illumination and viewing geometry improve as the pixel coarsens in comparison to the resolution of the DEM, the subpixel heterogeneity in the topography becomes more problematic. Perhaps mention that?

We have taken in account this insightful point when responding to the reviewer's comment on DEM further up. The following sentence added to the discussion (Line 638):

Indeed, as the resolution of the DEM increases compared to the satellite product, despite improvements in the estimationof illumination and viewing angles, the results become more sensitive to the sub-pixel heterogeneity.

Lines 625-630: Indeed the quality of the knowledge about the atmosphere is important, but so is sensor calibration. The paper is already long, so I avoid asking you to address the effect of uncertainty and drift in calibration, but at least mention it.

The following sentences have been added to the discussion (Line 669 in the updated manuscript):

Further sources of uncertainty can be linked to the satellite sensor itself. Although Sentinel-3 OLCI has an onboard calibration assembly performing periodical radiometric calibrations, unaccounted small changes in the stability of the sensors can occur. For example, it has been shown that Sentinel-3 OLCI overestimates radiance measurements over dark surfaces (Eumetsat, 2019). However, to the authors' knowledge no vicarious calibration studies have been performed over bright snow surfaces. Moreover, inter-sensor calibration differences should

be kept in mind when processing data from multiple identical satellites such as the Sentinel fleets (Clerc et al., 2020). Lastly, drift in the calibration of the sensor over time may lead to changes in the acquired data that in turn could be misinterpreted as physical processes (e.g. Casey et al., 2017).

Table 2. Indicate that Aspect is measured clockwise from North. This is the common convention, although it is inconsistent with a right-hand coordinate system. When I started working on topographic radiation problems, my go-to text was Physical Climatology (Sellers, 1965), with Aspect 0° to the South, positive east, negative west (as we use for longitude). Clockwise-from-North is most common, but not universal, hence the need to specify.

We fully agree with the reviewer's statement. This has caused misunderstandings within our research group in the past. The following sentence was added to the caption of Table 2:

Note that Aspect is measured clockwise, with 0° representing North.

REFERENCES:

Casey, K. A., Polashenski, C. M., Chen, J., and Tedesco, M.: Impact of MODIS sensor calibration updates on Greenland Ice Sheet surface reflectance and albedo trends, The Cryosphere, 11, 1781–1795, https://doi.org/10.5194/tc-11-1781-2017, 2017.

Clerc, S.; Donlon, C.; Borde, F.; Lamquin, N.; Hunt, S.E.; Smith, D.; McMillan, M.; Mittaz, J.; Woolliams, E.; Hammond, M.; Banks, C.; Moreau, T.; Picard, B.; Raynal, M.; Rieu, P.; Guérou, A. Benefits and Lessons Learned from the Sentinel-3 Tandem Phase. Remote Sens. 2020, 12, 2668.

Dozier, J. and Frew, J., "Rapid calculation of terrain parameters for radiation modeling from digital elevation data," in IEEE Transactions on Geoscience and Remote Sensing, vol. 28, no. 5, pp. 963-969, Sept. 1990, doi: 10.1109/36.58986.

Guyomarc'h, G., Bellot, H., Vionnet, V., Naaim-Bouvet, F., Déliot, Y., Fontaine, F., Puglièse, P., Nishimura, K., Durand, Y., and Naaim, M.: A meteorological and blowing snow data set (2000–2016) from a high-elevation alpine site (Col du Lac Blanc, France, 2720m a.s.l.), Earth Syst. Sci. Data, 11, 57–69, https://doi.org/10.5194/essd-11-57-2019, 2019.

Kokhanovsky, A., Lamare, M., Di Mauro, B., Picard, G., Arnaud, L., Dumont, M., Tuzet, F., Brockmann, C., and Box, J. E.: On the reflectance spectroscopy of snow, The Cryosphere, 12, 2371–2382, https://doi.org/10.5194/tc-12-2371-2018, 2018.

Larue, F., Picard, G., Arnaud, L., Ollivier, I., Delcourt, C., Lamare, M., Tuzet, F., Revuelto, J., and Dumont, M.: Snow albedo sensitivity to macroscopic surface roughness using a new ray-tracing model, The Cryosphere, 14, 1651–1672, https://doi.org/10.5194/tc-14-1651-2020, 2020.

Libois, Q., Picard, G., Dumont, M., Arnaud, L., Sergent, C., Pougatch, E., Vial, D. (2014). Experimental determination of the absorption enhancement parameter of snow. Journal of Glaciology, 60(222), 714-724. doi:10.3189/2014JoG14J015

Sirguey, Pascal, Renaud Mathieu, and Yves Arnaud. "Subpixel monitoring of the seasonal snow cover with MODIS at 250 m spatial resolution in the Southern Alps of New Zealand: Methodology and accuracy assessment." Remote Sensing of Environment 113.1 (2009): 160-181.

Sirguey, Pascal. Monitoring snow cover and modelling catchment discharge with remote sensing in the Upper Waitaki Basin, New Zealand. Diss. University of Otago, 2010.

Tuzet, F., Dumont, M., Arnaud, L., Voisin, D., Lamare, M., Larue, F., Revuelto, J., and Picard, G.: Influence of light-absorbing particles on snow spectral irradiance profiles, The Cryosphere, 13, 2169–2187, https://doi.org/10.5194/tc-13-2169-2019, 2019.